# Feedback-based sea level rise impact modelling for integrated assessment models with FRISIAv1.0

Lennart Ramme[1], Benjamin Blanz[2], Christopher Wells[3], Tony E. Wong[4], William Schoenberg[5,6], Chris Smith[7,8], and Chao Li[1]

[1]Max-Planck-Institute for Meteorology, Hamburg, Germany
[2]University of Hamburg, Hamburg, Germany
[3]School of Earth and Environment, University of Leeds, Leeds, United Kingdom
[4]School of Mathematical Sciences, Rochester Institute of Technology, Rochester, New York, USA
[5]University of Bergen, Bergen, Norway
[6]isee systems inc., Lebanon, New Hampshire, USA
[7]Vrije Universiteit Brussel, Brussels, Belgium
[8]International Institute for Applied Systems Analysis (IIASA), Laxenburg, Austria

**Correspondence:** Lennart Ramme (lennart.ramme@mpimet.mpg.de)

**Abstract.** Global warming is expected to lead to a substantial rise in coastal sea levels by the end of the century, which imposes future impacts and adaptation challenges on the coastal zone. Capturing the socio-economic costs of sea level rise (SLR) is therefore an important component of climate impacts in integrated assessment models (IAMs). However, there is a lack of process-based models of SLR impacts with a focus on global, time-varying dynamics. Current SLR impact models often
follow a cost-benefit analysis approach, fail to represent diverse pathways of SLR impacts, or do not include coastal adaptation. Here, we present the Feedback-based knowledge Repository for Integrated assessments of Sea level rise Impacts and Adaptation version 1.0 (FRISIAv1.0), a model designed for process-based, non-equilibrium IAMs that follows a system dynamics approach. FRISIA's SLR component is based on existing models of SLR, while its impact component is a substantially modified adaptation of the Coastal Impact and Adaptation Model (CIAM) for use in globally or regionally aggregated models.
While a reduced-feedback version of FRISIA approximately reproduces CIAM results, the integration of additional feedbacks in FRISIA leads to emerging new behaviours, such as a potential peak and decline in SLR-driven storm surge damages in the early 22$^{nd}$ century, due to economic feedbacks in the coastal zone. When coupling FRISIA to an IAM, global GDP is reduced by $1.5 - 6.2$ % ($17^{th} - 83^{rd}$ percentile range) under the mean SSP5-8.5 global-mean sea level rise from the IPCC's AR6 report (0.77 m by 2100) and no coastal adaptation, which is in the range of previous studies. The coupling of a diverse set of SLR
impact streams from FRISIA into a system dynamics IAM has the advantage of leading to a wide range of socio-economic consequences that go beyond just a reduction in global GDP, such as an effect on inflation. Our simulations highlight the benefits of accounting for dynamic coastal feedback and coupling diverse SLR impact and cost strains to IAMs, and showcase that FRISIAv1.0 is a useful tool for doing so.

# 1 Introduction

Sea level rise (SLR) is one of the main consequences of anthropogenic climate change that society will experience over the coming decades to centuries. Global mean sea level (GMSL) is likely to rise between 0.3 and 1.6 m by the end of this century, depending on the scenario and given uncertainties surrounding GMSL projections (Fox-Kemper et al., 2021). This magnitude of sea level change can have multiple and possibly severe consequences for coastal communities; it is therefore critical to account for the impacts of SLR when projecting the socio-economic evolution of the world in the 21[st] century and quantifying the effects of global climate change (e.g. Parrado et al., 2020; Brown et al., 2021; Bachner et al., 2022; Magnan et al., 2022; Hermans et al., 2023; Cortés Arbués et al., 2024).

Integrated assessment models (IAMs) are designed to produce possible future socio-economic scenarios under anthropogenic climate change. One of the use cases of IAMs is to generate the Shared Socioeconomic Pathways (SSPs) (Riahi et al., 2017), which in turn are used to drive Earth system model simulations (Eyring et al., 2016) and to inform the assessments of the Intergovernmental Panel on Climate Change (IPCC) (Masson-Delmotte et al., 2021). Hence, a crucial component of IAMs is a realistic representation of the connections between the human and the natural domain. The implemented connection from the human domain to the climate system arises primarily from greenhouse gas and aerosol emissions, as well as land-use changes, which are often simulated directly within the IAM (Popp et al., 2017). However, going into the other direction, the effects of climate change on the human system are less straightforward and often not process-based (Calvin and Bond-Lamberty, 2018; Donges et al., 2021; Dietz, 2024). For example, a damage function might simply reduce the gross domestic product (GDP) depending on the temperature anomaly as originally proposed by Nordhaus (1977). Consequently, IAMs are often criticized for missing internal feedbacks and heterogeneity (Donges et al., 2017; Calvin and Bond-Lamberty, 2018; Keppo et al., 2021).

Within the typical IAM structure, the impacts of SLR are often represented by simple implementations, if included at all, as we describe in more detail in Sect. 2. Currently, more comprehensive SLR impact and adaptation formulations can be found in IAMs that are computable general equilibrium (CGE) models or those that employ cost-benefit analysis approaches (Waldhoff et al., 2014; Yumashev, 2020; Rennert et al., 2022). These models try to find cost-optimal pathways or maximize welfare by solving a given set of equations (Keppo et al., 2021). However, the realm of IAMs also encompasses more process-based models (Weyant, 2017), and a subset of those simulates forward on a time step basis without solving for cost-optimal pathways (Keppo et al., 2021). This modelling approach is part of the field of system dynamics, which focuses on the understanding of nonlinear complex systems by emphasising the feedback between components (Forrester, 1961; Sterman, 2000). Hence, for simplicity, we refer to these types of models as dynamic IAMs. Dynamic IAMs typically value dynamic complexity, that is, the interconnectedness of processes and model components, over process detail. This has the advantage of higher computational efficiency, increased transparency and a better representation of time-dependent behaviour, whole-system feedback and uncertainty (Weyant, 2017; Donges et al., 2017). However, there is no model for SLR impacts and adaptation that specifically meets these criteria for the use in dynamic IAMs. This is where our study comes in.

We present the Feedback-based knowledge Repository for Integrated assessments of Sea level rise Impacts and Adaptation (FRISIA) version 1.0, a newly developed model that aims to improve the representation of SLR impacts in global or regional

dynamic IAMs. FRISIA's focus is on incorporating socio-economic feedback within the coastal zone that is missing in other SLR impact models and to provide a wide range of output streams that can be coupled to different parts of an IAM. Incorporating SLR impacts into IAMs with FRISIA will therefore improve the realism of the feedback from the natural to the human domain in these models.

## 2  Sea level rise impacts and integrated assessment modelling

Rising sea levels lead to increased coastal erosion, the intrusion of saltwater into coastal aquifers, and increases in the frequency and severity of extreme storm surges, leading to increased flood damages (Cazenave and Cozannet, 2014). At the same time, SLR is a slow process, happening on a time scale that is longer than the time it typically takes for capital and infrastructure to depreciate, so that actual damages from gradual flooding might be smaller (Desmet et al., 2021). Nevertheless, if not protected, low-lying areas within the coastal zone may become inundated so that people and economic assets are forced to migrate away from the coast (Lincke and Hinkel, 2021; Duijndam et al., 2022). Furthermore, there are multiple ways in which the coastal community can adapt to SLR, which range from raising flood protection, flood-proofing or raising existing structures, to renourishing of beaches or other forms of soft protection, and the planned retreat of assets and people from the coast (Oppenheimer et al., 2019; Tiggeloven et al., 2020). However, determining the optimal adaptation strategy is highly dependent upon the local geography, the socio-economic boundary conditions and potentially cultural heritage (Nazarnia et al., 2020; Bongarts Lebbe et al., 2021). Therefore, IAMs that model the global impacts of SLR must in principle rely on models that divide the coast into a large number of segments within which the geographical and socio-economic properties are approximately similar.

Most commonly used for breaking up the coast in smaller segments is the Dynamic Interactive Vulnerability Assessment (DIVA) database, which assesses damages based on the local conditions of 12,148 distinct coastal segments (Vafeidis et al., 2008; Hinkel and Klein, 2009; Hinkel et al., 2014; Lincke and Hinkel, 2018). The open-source Coastal Impact and Adaptation Model (CIAM) builds on the DIVA database and was designed to determine local SLR impacts and the optimal least-cost adaptation strategy to quantify the global economic impacts of SLR (Diaz, 2016). Expanding on the scope of the DIVA database, Depsky et al. (2023) have presented the DSCIM-Coastal platform, which provides a similar database as DIVA, but with major updates for all types of included information, which is named the Sea Level Impacts Input Dataset by Elevation, Region and Scenarios (SLIIDERS). DSCIM-Coastal also provides an updated python version of CIAM (pyCIAM) and is available as open-source software. Models like CIAM and DSCIM-Coastal are suitable tools for quantifying the global costs and damages associated with SLR and characterizing uncertainty in the economically-efficient adaptation strategies across a wide spatial domain (Wong et al., 2022; Depsky et al., 2023). For example, both models were included in a recent accounting of the social-cost of carbon (SCC) (Rennert et al., 2022; ImpactLab, 2023). Rennert et al. (2022) found that accounting for impacts of SLR only adds around $1 - 2\%$ to the total SCC. However, the model assumed optimal adaptation strategies, and as SLR damages occur far into the future, the costs associated with the impacts of SLR are strongly discounted (a discount rate of 2% is used in that study). Similar to the GIVE model used in Rennert et al. (2022), the IMAGE model also tracks local SLR and its impacts

based on the DIVA database (Stehfest et al., 2014). Nevertheless, the complexity of CIAM or DSCIM-Coastal precludes their use within IAMs that require a computationally more efficient model. Consequently, the representation of sea level rise impacts is much simpler in most other existing IAMs.

Some IAMs, such as the highly aggregated DICE model (Nordhaus, 2017; Barrage and Nordhaus, 2024), the more process-based REMIND (Baumstark et al., 2021) model, or the system dynamics model FeliX (Ye et al., 2024) do not explicitly track SLR, instead incorporating damage functions that are only a function of the global mean temperature. While this may be a tenuous but defensible assumption in scenarios with continuous warming, this approach becomes problematic in scenarios of mitigated global warming. In this case the rate of global warming slows down or even reverses, but global mean sea level continues to rise in the long term due to inertia and hysteresis effects of the ocean and the ice sheets (Li et al., 2020). Furthermore, the contribution of land water storage (LWS) changes to GMSL rise is more directly a function of socio-economic conditions and not of temperature (Reager et al., 2016). Going a bit further, the PAGE model explicitly tracks GMSL as a function of the global mean temperature anomaly and calculates damages that account for local per capita income and adaptation (Moore et al., 2018; Yumashev, 2020). A more comprehensive formulation of SLR damages and adaptation in an integrated assessment model is implemented in the FUND model (Tol, 2007). In FUND, adaptation to SLR is represented based on a cost-benefit analysis, accounting explicitly for the costs of wetland and dryland loss, coastal protection and migration, and including the effect of expected future SLR.

In summary, many IAMs do not specifically account for SLR impacts (e.g., DICE, REMIND, FeliX) and those that do are often aggregated computable general equilibrium (CGE) models (PAGE, FUND). Only the GIVE and IMAGE models incorporate more comprehensive SLR impact formulations, but their high spatial resolution makes them difficult to use in more aggregated IAMs that focus on dynamic complexity instead of process-detail. This means that, while all the above-mentioned models are well established and have their use cases, there is a lack of a process-based SLR impact and adaptation model for use in dynamic IAMs. Nevertheless, also CGE models and other types of IAMs can benefit from an improved and computationally efficient implementation of SLR impacts and adaptation.

The FRISIA model version 1.0 that we present in this manuscript is a model for SLR and the corresponding impacts and adaptation to it, which fills the aforementioned gap. The model is named after the Frisia region along the southern coast of the North Sea, which has historically experienced major SLR and storm surge damages (Bungenstock et al., 2021; Von Storch and Woth, 2008). The name was also chosen in reference to the new integrated assessment model FRIDA (Schoenberg et al., 2025), for which FRISIA was initially developed, even though FRISIA is an independent model that can be run uncoupled or coupled to other IAMs. FRIDA follows the same system dynamics modelling approach as FRISIA, making it the natural home of our model. As a system dynamics model, in FRISIA we try to capture the inter-relatedness of SLR-related processes, but at the same time we sacrifice process-detail and resolution for dynamic complexity. Because we also value transparency and accessibility, FRISIA is developed in Python and made available in an open source GitHub repository (see Code Availability section). In Sect. 3 we describe the SLR model and FRISIAv1.0 formulation of SLR impacts and adaptation. In Sect. 4 and Sect. 5 we then present results of our model when used in an uncoupled mode, whereas in Sect. 6 we couple the model to FRIDA and explore the resulting impacts.

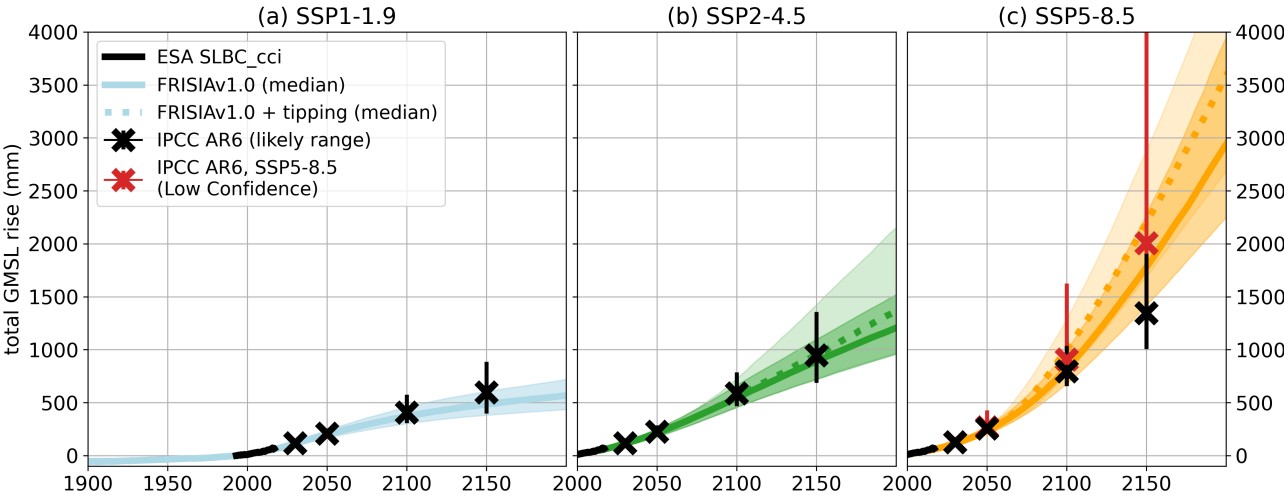

**Figure 1.** The total GMSL rise as projected with FRISIA for SSP1-1.9 (a), SSP2-4.5 (b) and SSP5-8.5 (c) compared to data of the European Space Agency (ESA) Sea Level Budget Closure - Climate Change Initiative (SLBC_CCI) project (Horwath et al., 2021, black curve) and aligned to the year 1993. The solid and dashed curves are the cases with deactivated and activated high impact behaviour parameterization in FRISIA, respectively. Also given are the median estimates for 2030, 2050, 2100 and 2150 from AR6 (Fox-Kemper et al., 2021). We additionally present the "Low Confidence" scenario as the red error bars for SSP5-8.5, which includes uncertain high impact processes. Given error bars are the "likely ranges" from Table 9.9 in Fox-Kemper et al. (2021), which represent the range with a 66% likelihood of occurrence in the IPCC's calibrated language. Correspondingly, the median and the $17^{th} - 83^{rd}$ percentile ranges for the data from FRISIA are shown as lines and shaded area, respectively.

## 3 Model description

### 3.1 Sea level rise model formulation and projections

The first step to address the impacts of SLR is to calculate the magnitude of GMSL rise under global warming. There are many models that are designed to quantify GMSL within the framework of IAMs or similar use cases (Rahmstorf, 2007; Nauels et al., 2017; Wong et al., 2017). The simplest possible choice that could easily be used in an IAM would be the semi-empirical approach of Rahmstorf (2007), which relates the annual change in GMSL linearly to the global mean surface air temperature (GSAT) anomaly with respect to the pre-industrial period. However, while this fits GMSL observational data and leads to projections for 2100 that are approximately in line with IPCC estimates (Fox-Kemper et al., 2021), the approach neglects the non-linear nature of many of the processes contributing to changes in GMSL, and it is not certain that the relationship remains valid in the more distant future. Furthermore, the same rate of GMSL rise can translate into different rates of local SLR, due to the locally varying impacts of the different SLR contributions (Palmer et al., 2020; Fox-Kemper et al., 2021). Hence, as

we value dynamic complexity and process-based relationships, and since FRISIA aims at also being suitable for regionalised IAMs, we explicitly model the five most important contributions to GMSL rise: thermal expansion of the ocean, changes in land water storage, the melting of mountain glaciers, the Antarctic ice sheet (AntIS) and the Greenland ice sheet (GrIS). Again, all these processes are already represented in existing models of GMSL rise, and we highlight that the individual components of FRISIA are largely based on the two existing models from the BRICK (Wong et al., 2017) and MAGICC (Nauels et al., 2017) modelling frameworks. Nevertheless, as small modifications were made to some of the parameterisations, and in order to provide comprehensive model documentation, we describe the formulation of the individual GMSL rise contributions in detail in the appendix Sect. A.

We drive the SLR model with time series of GSAT and ocean heat content (OHC) anomalies from FaIR v2.1.1 (Smith et al., 2018; Leach et al., 2021), which is a well-established climate emulator that reproduces the behaviour of the CMIP6 ensemble and provides a range of possible future projections. We use a calibration of FaIR (v1.1.0) that is constrained to observed time series of global mean surface temperature for 1850–2019 and OHC change for 1971–2018 along with IPCC assessments of the uncertainty distributions of equilibrium climate sensitivity, transient climate response, present-day aerosol forcing and present-day $CO_2$ concentrations (Smith et al., 2024), driven with emissions scenarios from the IPCC Sixth Assessment Report and containing 1001 ensemble members. The GSAT anomaly can be used as the driver of the three GMSL rise contributions that are related to the melting of ice. Furthermore, OHC change is required to calculate the thermosteric component of GMSL rise, while the land water storage component is driven by socio-economic variables. For the latter, when running uncoupled to an IAM, we use global population data and projections taken from the IIASA SSP database (Riahi et al., 2017; Samir and Lutz, 2017). When running coupled to an IAM, all these variables are often readily available endogenously.

Modelling GMSL rise comes with a large range of structural and parametric uncertainties at the level of the individual SLR processes, which add to the uncertainty in the variables that drive GMSL rise. Therefore, model calibration is essential to produce realistic outcomes that fit with historical observations and match future projections from more detailed models (Fox-Kemper et al., 2021). Two different calibration targets are therefore used here for calibrating our model. We make sure that observational data for the individual contributions to GMSL rise are matched, which we take from Horwath et al. (2021). At the same time, we calibrate to the GMSL projections from Chapter 9 of the IPCC's Working Group 1 contribution to the Sixth Assessment Report (AR6, Fox-Kemper et al., 2021) for three scenarios (SSP1-1.9, SSP2-4.5, SSP5-8.5), covering the maximum range of plausible future pathways represented by the SSPs.

Given the large uncertainties in the parameter values, as well as the structure itself, we value simplicity over detail during the calibration process. Therefore, for each of the five components of GMSL rise, we define a "scale factor" that can be varied between 0 and 1, which respectively correspond to the minimum and maximum contribution of the individual components. These factors are varied randomly and independently for each member of the FaIR input ensemble, so that the resulting time series of GMSL rise include uncertainty from both climate projections and SLR responses. Inside the SLR formulation for each component, this factor is used to scale a set of one to four parameters within a predefined range simultaneously. The calibrated range of the uncertainty parameters is given in the appendix Table A1. While this approach does not cover the full parametric uncertainty, it makes sure that parameter combinations leading to unrealistic outcomes are avoided, and it is sufficient to cover

the uncertainty ranges in AR6. These parameter ranges are defined during the calibration process and are described in appendix Sect. A.

In Fig. 1, we present FRISIA projections of the total GMSL rise, calculated as the sum of the five individual contributions. The internal model parameters are hand-calibrated to match the projections of the individual contributions for the year 2100 presented in AR6 (Table 9.9 in Fox-Kemper et al., 2021), but not to specifically match those of the total GMSL rise shown in Fig. 1. Nevertheless, the projected total GMSL rise fits well with estimated median values and uncertainty ranges until 2100. However, in the 22nd century, our model projects a larger sensitivity to the long-term evolution of the GSAT anomaly, leading to values of GMSL rise that are slightly below the AR6 estimates for 2150 in SSP1-1.9, while they are above the estimates for SSP5-8.5 (excluding the "Low Confidence" scenario of very high SLR). This is not because of the temperature time series used, but a consequence of the parametrisations of the model. For the intermediate scenario SSP2-4.5 the fit is good even for the year 2150, and we note that IAMs rarely simulate time horizons that go beyond the year 2100, due to the sizeable uncertainties in the socioeconomic evolution on such long time scales.

## 3.2 SLR impacts and adaptation model

The impacts and adaptation component of FRISIA is designed on the basis of the global coastal impact and adaptation model CIAM (Diaz, 2016; Wong et al., 2022). The choice of CIAM and hence the DIVA database over DSCIM-Coastal was made because of initial data availability, but we discuss a potential future use of DSCIM-Coastal in the discussion section (Sect. 7).

In contrast to CIAM, FRISIAv1.0 is set up as a global or regionally aggregated model, because it is designed for use in IAMs, in which computational speed and interactive coupling are valued. FRISIAv1.0 is designed for dynamic IAMs that integrate forward in time. Therefore, the model deviates from CIAM in that it uses a comparably small time step of one year, and only calculates the annual costs over this time step. This means that FRISIA does not calculate the costs of all possible adaptation strategies and then chooses the strategy that minimises the costs over a given time horizon. Instead, the model simply calculates the evolution of coastal assets and population, as well as the coastal protection level and annual damages and costs for a user-defined adaptation strategy. While this approach does not inform the user about the cost-optimal strategy, which would not be informative on a coarse aggregation level and often does not represent reality, there are some key advantages. First, the chosen approach allows for an easier analysis of feedbacks within a given adaptation strategy. Second, it gives an improved representation of the annual evolution of impacts. Third, it enables an integration of an endogenous framework of decision making in the future, which takes into account more aspects than just cost-optimisation.

In this section we discuss all the components of the impacts and adaptation model of FRISIAv1.0 in detail. A schematic of the model is shown in Fig. 2, which shows the logical flow of information between all components. We organize the model description based on this. In the first subsection we describe how FRISIA generally uses information from CIAM and the DIVA database, as well as how other input data is used to drive the model. We then continue with a description of how we derive coastal flooding and the exposure of assets and people to SLR damages from time series of GMSL and its subcomponents. The remaining three subsections then describe the technical implementation of how flood protection, as well as the dynamics of coastal assets and coastal population are modelled (yellow boxes in Fig. 2).

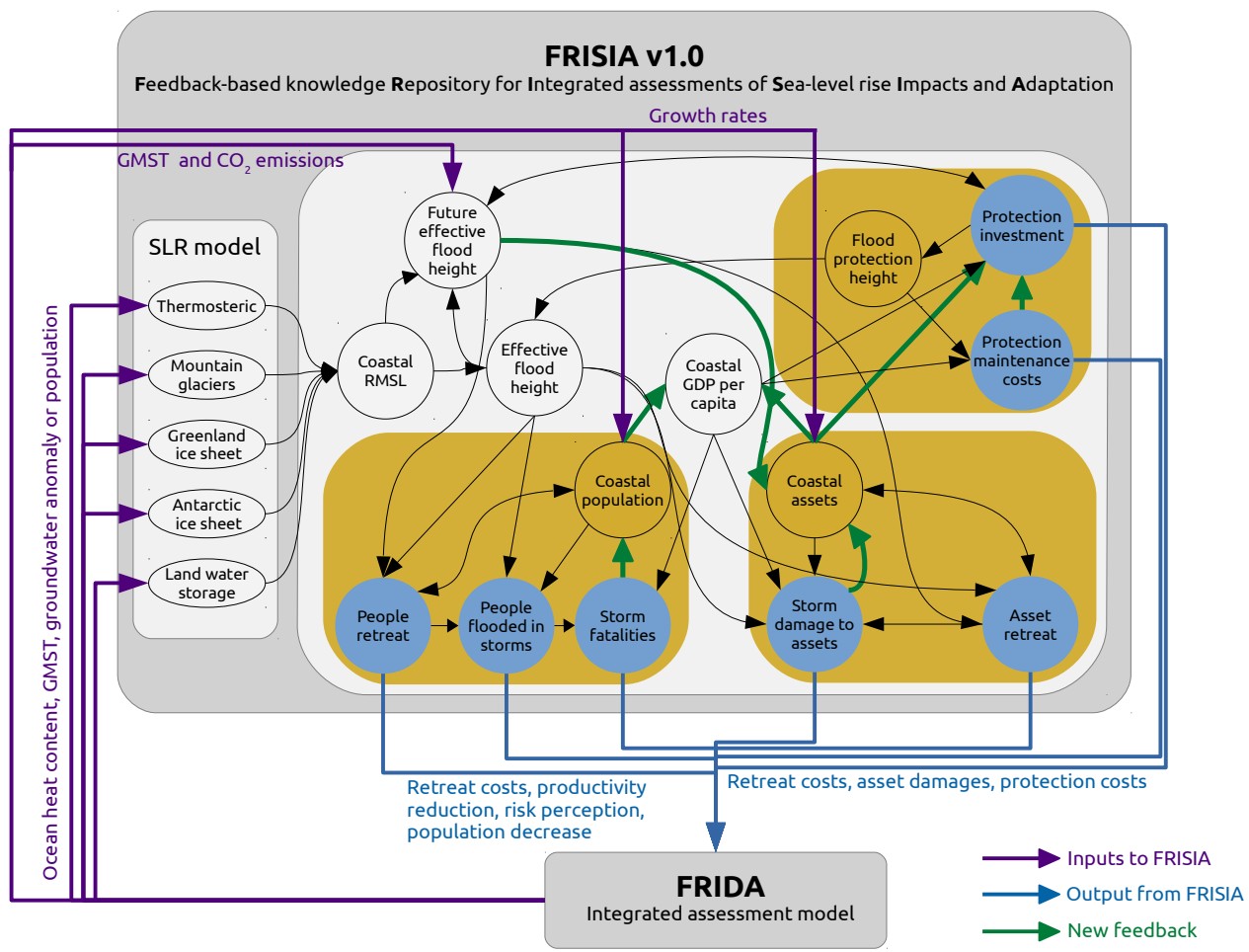

**Figure 2.** Schematic overview of the FRISIA model and how it is coupled to FRIDA (see Sect. 6). In an uncoupled version, the inputs to FRISIA (purple arrows) come from external datasets and the outputs (blue arrows) do not feed back onto the model that creates the input. The green arrows represent new connections that are made in FRISIA compared to CIAM, and which are turned off in Sect. 4.2. Blue circles are used for variables that represent SLR impacts.

### 3.2.1 Model inputs

We aggregate data from the DIVA database in order to include coastal information in FRISIA. In the current version there are three types of aggregation: (1) a global setup, including all segments from the DIVA database; (2) a bipolar setup, where these segments are separated into those with a dike height that is higher than the height of a storm surge with a return period of 1000 years ("well protected" segments) and those where this is not the case ("less protected" segments); and (3) a regional setup, where the DIVA segments are separated into the seven World Bank regions (World Bank Group, 2025). These data are stored in input files in the repository and loaded into the model at initialisation, depending on the chosen aggregation type. For each aggregation type we store the coastline length, the initial flood protection height, coastal assets and population. The population data is taken from the DIVA dataset directly, and asset values are assumed to be proportional to coastal GDP data (see Sect. 3.2.4), which is calculated using the DIVA population and country-level GDP per capita data (Riahi et al., 2017; Cuaresma, 2017).

FRISIA also reads in SLR weights for each coastal zone. These weights are taken from Slangen et al. (2014) and aggregated by calculating a population-weighted mean over all the segments within each coastal zone. There are weights for the individual SLR components, whereby those for thermal expansion and land water storage are always equal to one. When the respective time series of the SLR components are provided by the user, FRISIA will use these weights to calculate the local SLR for each coastal zone. There is an additional weight for the total GMSL rise, based on fitting the local SLR in each coastal zone to GMSL rise in the SSP5-8.5 scenario. This represents a simplified approach that is only used in case there is only total GMSL rise as input to the model, or when calculating expected SLR in the future, as described in Sect. 3.2.3.

The overall approach of FRISIAv1.0 is to use the same model formulations as CIAM, where appropriate in a highly aggregated model like FRISIA. This is the case for the general flood protection construction and maintenance formulations, which are identical to those in CIAM, except for the calculation of the desired protection level and the available amount of money for investment (Sect. 3.2.3). Also the calculations of land values, opportunity costs, and relocation costs use the same formulations as CIAM. Where FRISIA uses the same model formulations as CIAM, it also uses the same parameter values, but with FRISIA we added approximated ranges of uncertainty for all of the parameters, including those of the new model formulations unique to FRISIA (see appendix Table A3).

For the baseline growth of asset and population stocks, the offline version of FRISIA uses the same input data from the SSP database as CIAM (Riahi et al., 2017; Cuaresma, 2017). This means that coastal population and GDP per capita growth rates are based on country-level rates, which are extended with constant values beyond 2100 (Fig. A7), leading to visible kinks in that year for some variables. These are applied to each coastal segments, before aggregating the information into GDP and population data for each coastal zone. When using FRISIA coupled to an IAM, the IAM should provide the growth rates directly.

We have now discussed the input data for coastal assets and population, how coastal SLR is calculated from time series of GMSL rise and for which parts of FRISIA CIAM formulations are applied directly. However, this is not yet enough local information in order to properly calculate storm surge damages and inundation of people and assets, i.e. coastal flooding.

### 3.2.2 Modelling coastal flooding

In order to model coastal flooding, FRISIA also makes use of processed data from the CIAM model, which is ultimately based on the DIVA database, but it was not possible to use the CIAM formulations for this directly. This is because FRISIA would aggregate segment level data, but probability distributions of storm surge levels are not meaningful at a high level of aggregation. Therefore, our approach is to sum up the generally susceptible, the annually exposed, and the fully inundated coastal assets and population from each segment in a coastal zone respectively, expressed as fractions of the overall coastal stocks. This is done in a pre-processing step that calculates these fractions for each GMSL data point from the SSP5-8.5 scenario until 2200, in order to find relationships between these fractions and a maximum possible range of GMSL values that is relevant for FRISIA. We use a high emissions scenario, rather than a simple array of possible GMSL values, in order to attain realistic values for local SLR for, both, the aggregated coastal zone and each coastal segment. This means for a given "regional-mean" sea level (RMSL) rise within a coastal zone, the corresponding local SLR is used to calculate the above mentioned fractions within a segment. This is where the DIVA database is used again. DIVA provides 15 segment-level area parameters that can be used in the following functional form, as in the CIAM model, to calculate the inundated area for a given increase in local sea level:

$$
\begin{aligned}
area_{ind} =& a_0 \cdot \max(0, \min(0.5, sl)) + \left(\frac{a_0 + a1}{2}\right) \cdot \max(0, \min(1, sl - 0.5)) \\
&+ a_1 \cdot \max(0, \min(0.5, sl - 1.5)) + \sum_{i=2}^{13} a_i \cdot \max(0, \min(1, sl - i)) \\
&+ a_{14} \cdot \max(0, sl - 14)
\end{aligned}
\tag{1}
$$

$sl$ in this equation is the local SLR for each segment, when calculating inundated fractions, but it is local SLR plus the height of a 1000-year return period storm surge in this segment, when calculating the generally susceptible area. We add this surge height, which is the highest surge level in DIVA, to get the maximum area that could possible be susceptible to storm surges. For both, the calculated area in each segment is set to zero, if the local SLR is smaller than the flood protection height $H$ in the segment, so that no flooding would occur. The inundated area can then be combined with the segments' population density and GDP per capita data to calculate the fully inundated fractions of people and assets for each level of local SLR in each segment, assuming an even distribution within the segment.

The same approach is used for calculating the people and asset fractions that are annually exposed to temporary flooding, that is, storm surges. For this, we use the following formulation of the CIAM model that calculates annual exposure area based on segment-level return periods and local SLR for each segment ($sl$):

$$
area_{exp} = \frac{\sigma_0 + c \cdot \max(0, sl)}{1 + A \cdot \exp(B \cdot \max(0, H - sl))}.
\tag{2}
$$

The parameters $A$, $B$, $c$ and $\sigma_0$ are available for each segment from the CIAM model repository. Again, population density and GDP per capita data are used to turn area into exposed population and assets.

The segment-level data for annual exposure and general susceptibility to storm surges, as well as inundated assets, people and land are then summed up over all segments that constitute a coastal zone in FRISIA and divided by the overall amount

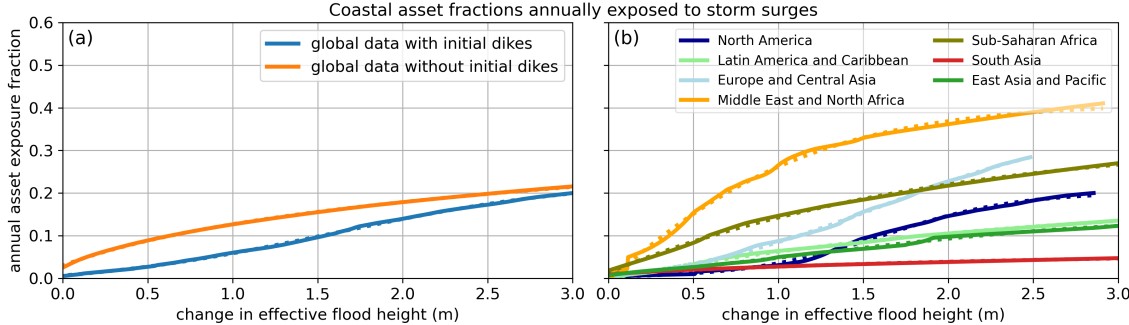

**Figure 3.** The fraction of the original asset distribution that is annually exposed to storm surges, $F_{\text{exp}}^{\text{A}}$, as a function of changes in the effective flood height $S$. (a) Data from the global aggregation setup for the cases of including and not including the initial dike height from the DIVA dataset.(b) Data of the regional aggregation setup, when including the initial dikes. The solid lines are the aggregated data from the CIAM exposure estimation and the dotted lines are the fits used to represent this information in FRISIA.

of people and assets in that coastal zone (except for inundated area, which is used in absolute terms). For each coastal zone, these variables can then be expressed as a function of the mean "effective flood height" in that coastal zone, as it is shown in Fig. 3 for the annual asset exposure to storm surge damages. The effective flood height $S$ is thereby simply the mismatch between the current RMSL anomaly in that coastal zone and the increase of the flood protection height since the beginning of the integration at $t_0$:

$$S(t) = (\text{RMSL}(t) - \text{RMSL}(t_0)) - (H(t) - H(t_0)),\tag{3}$$

i.e. it is a measure of how much the protection level changed relative to the initial level. This follows the simple assumption that an increase in flood protection height can fully balance any equivalent level of RMSL rise. We note that the current version of FRISIAv1.0 might therefore underestimate damages in mixed scenarios, in which flood protection height is increased, but this increase cannot keep up with RMSL rise. This does not affect the standard experiments presented in this manuscript.

As the aggregation of data is done in a pre-processing step, we continue to express the aggregated data in a functional form, in order to be able to load and use them in FRISIA, as shown exemplary in Fig. 3 for coastal asset exposure. Logistic functions are used to parametrize the data, if the initial dike height from the DIVA dataset is taken into account (blue line in Fig. 3 (a)). A logarithmic function is used in the counterfactual reference scenarios without initial flood protection (orange line in Fig. 3 (a)) that we use in Sect. 4.2 for comparison to CIAM. The functional representation of storm surge exposure is in fact the

main determinant of how future damages of SLR evolve in the case of no adaptation. From the blue line in Fig. 3 (a) it can be seen that the global annual storm surge exposure of assets is roughly a linear function of the effective flood height, and hence GMSL rise. However, Fig. 3 (b) shows that there is a big spread in how the exposure in different regions responds to RMSL rise, which means that simple linear relationships between GMSL rise and coastal damages will fail when extrapolated beyond a global aggregation level.

As a consequence of our highly aggregated modelling approach, our representation of coastal flooding is a major simplification of reality, as it does not incorporate any detailed information on coastal morphology, shoreline type or local wave heights, which would be necessary to explicitly model impacts of SLR at the local scale. This is a limitation of our modelling approach, which we discuss further in Sect. 7, but highlight here that the same simplifications are made in CIAM, which is more disaggregated than FRISIA, and that representations of SLR impacts in IAMs are typically far less complex.

### 3.2.3  Coastal protection

We follow the CIAM model in its formulation of coastal protection and track an aggregated flood protection height $H$ as the only form of generalised protection against rising sea levels. This simple approach is necessary, because the most suitable form of flood protection is highly dependent on local boundary conditions, which are not tracked in the model. The initial flood protection height $H_0$ is taken as the mean dike height from the DIVA data set, averaged globally or for each region depending on the aggregation level. The total cost of coastal protection is comprised of three components: the cost of raising flood defenses, maintaining them, and the opportunity cost of the land used for building them.

The cost to raise flood protection further by $\Delta H$ is calculated, as in CIAM, as

$$\text{Cost}_{\text{protect,build}} = c_{\text{construct}} \cdot L \cdot \left( (H + \Delta H)^2 - H^2 \right), \tag{4}$$

where $L$ is the flood protection length. The construction cost $c_{\text{construct}}$ is defined as the product of a reference construction cost of 0.00602 (0.005$-$0.007, upper and lower parameter bounds) billion USD 2010 (from here on b\$) km$^{-1}$ m$^{-2}$ as in CIAM, and a construction cost index (cci) that depends on GDP per capita in the coastal zone. The cci formulation is fitted with a linear function to the cci data from the CIAM repository (taken from the World Bank Group (2017) International Comparison Program (ICP); updated compared to Wong et al. (2022)), and country-level estimates of GDP per capita. Like in CIAM, values for the cci are limited between 0.5 and 2.5, to avoid unrealistic construction costs (Wong et al., 2022).

Maintenance costs for the raised flood protection are tracked as

$$\text{Cost}_{\text{protect,maintain}} = f_{\text{maintain}} \cdot c_{\text{construct}} \cdot L \cdot H, \tag{5}$$

with $f_{\text{maintain}} = 0.02$ (0.015$-$0.03) being the fraction of construction costs that are required annually for maintenance (Diaz, 2016). When calculating the total SLR adaptation costs, we subtract from the total costs of maintenance the costs that would occur if the flood protection height was not changed from its initial height $H_0$. This is done to track only those adaptation costs that are induced by SLR and not by increased construction costs. We point out that in our approach the simulated construction of flood protection only tracks what would be constructed in response to SLR. Hence, the assumption is that the initial protection levels at the start of the integration period are maintained; we follow this approach because we are only interested in the additional costs attributable to SLR under climate change.

A third cost component is the opportunity cost of using the land for flood protection, calculated as in CIAM

$$\text{Cost}_{\text{protect,land}} = f_{\text{land}} \cdot lv \cdot L \cdot \Delta H \cdot 1.7, \tag{6}$$

where the factor 1.7 comes from the ratio of sea wall width over height (Diaz, 2016). The land value $lv$ is defined using an initial average land value of 0.005376 $(0.005-0.006)$ b\$ $km^{-2}$ (Darwin, 1995; Wong et al., 2022). The land value grows over time, depending on demand and willingness to pay, which can be approximated as a function of coastal GDP per capita and population (Yohe et al., 1999). This is how it is applied in the CIAM model (Diaz, 2016), and we follow this approach. The

320 annual opportunity cost rate of land is $f_{land} = 0.04 \pm 0.01$ $yr^{-1}$ (Wong et al., 2022).

There is some loss of information going from CIAM to FRISIA in this instance, because of the necessary aggregation across all segments. The most important issue is that the required flood protection investment is quadratic in height. Using the mean flood protection height over all segments in a coastal zone therefore means that calculated costs are not exactly equal to those in CIAM, given that everything else would be the same. However, most other formulations are linear, and the fact that there is

325 only one type of flood protection, the cost of which are proportional in coastline length, means that the CIAM formulations are generally also useful in FRISIA.

The annual amount of money that is invested into flood protection is derived from the theoretical investment needed to restore and maintain the initial protection level under current and expected SLR over the next 50 years. We choose a time horizon of 50 years, because, on the one hand, this gives enough time for building sufficient protection, while, on the other hand, there is

330 still a relative small uncertainty in SLR projections over 50 years compared to the high uncertainty over longer time scales (see Fig. 1). In general, this follows the assumption that coastal zones, which adapt to SLR by building flood protection, will do so not just in response to experiencing changes in sea level, but using their knowledge about future SLR.

However, as the original purpose of FRISIA is to be run coupled to an IAM, which interactively provides the drivers for annual rates of GMSL rise, it will not always have access to pre-computed time series that provide the actually modelled

GMSL rise in 50 years. Instead, we calculate the expected GMSL rise over the next 50 years, $SLR_{50}$, using a fitted linear relationship depending on the current GSAT anomaly and $CO_2$ emissions. This relationship was derived using the model of GMSL presented in this manuscript, and it gives an estimate that is very close to the actually modelled GMSL rise in 50 years (Fig. A8). While this relationship was fitted using data of GMSL, the expected RMSL for each coastal zone can be calculated using the pre-computed weights for each coastal zone, which is provided as pre-processed input as described in Sect. 3.2.1.

We note that this assumes that the relative contribution of the individual SLR processes stays approximately the same as in the projection used for the fit (SSP5-8.5). Furthermore, the fit was purposely done without activating the possibility of high impact processes on the Greenland and Antarctic ice sheet contributing to GMSL rise, indicating that the expected GMSL rise will underestimate the actual GMSL rise in 50 years if the high impact switch is activated when running the model.

The expected SLR over the next 50 years is then used together with the current effective flood height, $S$, to compute the

345 required additional flood protection height. The required investment to maintain the initial protection level then is

$$I_{50} = c_{construct} \cdot L \cdot \left( (SLR_{50} + S + H)^2 - H^2 \right). \tag{7}$$

The actual annual investment in flood protection $I$ is further modified from $I_{50}$, first using a scale parameter $W_{protect}$ with values between 0 and 1, which can be interpreted as the political will to invest in flood protection. Next, the total investment is divided by the time it takes to build flood protection $\tau_{protect}$, which we assume to be 10 $(5-25)$ years, as in CIAM (Wong et al.,

2022). Finally, we calculate the maximum amount of money available annually for flood protection as a constant fraction of the coastal GDP, $f_{\text{invest}}$, which has to also cover flood protection maintenance costs. The actual annual investment is therefore

$$I = \text{MIN} \left( W_{\text{protect}} \cdot \frac{I_{50}}{\tau_{\text{protect}}}, I_{\text{available}} \right), \text{ with} \tag{8}$$

$$I_{\text{available}} = \text{MAX} \left( 0, f_{\text{invest}} \cdot \text{GDP}_{\text{coast}} - \text{Cost}_{\text{protect,maintain}} \right), \tag{9}$$

and it is directly transformed into an increased flood protection height in the next time step. As CIAM does not include such a
355 limit on flood protection spending, we deactivate this limit in our comparison to CIAM results as well. The limit is activated during the respective feedback analysis and the simulations coupled to an IAM.

### 3.2.4 Coastal assets

FRISIA tracks a stock of coastal assets that is initialised by calculating coastal GDP from population density (taken from DIVA) and country-level GDP per capita data, following the SSP scenarios (Riahi et al., 2017; Cuaresma, 2017). We further follow
CIAM (Diaz, 2016) and assume a constant asset to GDP ratio of 3 (Nordhaus, 2010). This means that asset stocks in FRISIA are a broadly defined representation of the total value of past investments that produce economic output, after accounting for depreciation. This includes, but is not limited to, residential and non-residential buildings, infrastructure, equipment and machinery (Samuelson and Nordhaus, 2001). There is no disaggregation of assets and no relative weighting by their vulnerability to SLR. Coastal assets can be increased or reduced via four different pathways.

1. Asset growth: Asset values grow at a rate $r_{\text{A}}$ that is either the coastal GDP growth rate, assuming a constant future asset to GDP ratio (Nordhaus, 2010; Diaz, 2016), or directly using an asset growth rate, if this is provided by the IAM. In addition, in FRISIA the asset growth can be reduced if the expected effective flood height in 50 years, $S_{50}$, is growing. This reflects reduced investment and asset value reductions under increased future storm damage risk. A feedback of SLR exposure on asset prices has been reported in some studies in terms of a discount on real estate prices in exposed areas (Bin et al., 2011;
McAlpine and Porter, 2018; Bernstein et al., 2019; Keys and Mulder, 2020; Tyndall, 2023); we refer to Contat et al. (2024) for a recent review. We here assume that this effect will grow with increased exposure to SLR, and that it affects all types of assets. We calculate the expected effective flood height using the expected SLR over the next 50 years and the potential increase of the flood protection height $\Delta H_{50}$, which is the height increase under a continuation of the current annual investment in flood protection. Together this gives

$$S_{50} = S + \text{SLR}_{50} - \Delta H_{50} \tag{10}$$

We then use the expected effective flood height to reduce asset growth by the likelihood of investment

$$\rho = 1 - \text{MAX} \left( 0, \frac{S_{50}}{S_{50,\text{halfpoint}} + S_{50}} \right), \tag{11}$$

where $S_{50,\text{halfpoint}}$ is the expected future effective flood height at which only half of the investment would be made. We use $S_{50,\text{halfpoint}}$ as the uncertainty parameter in this formulation with a large range of allowed values (0.5–3.0 m), because of the

large uncertainty around the effect of SLR on asset prices (Contat et al., 2024). If $\rho < \rho_{\text{threshold}} = 0.95 \pm 0.05$, the change of asset values $A$ due to GDP growth is

$$\left( \frac{\mathrm{d}A}{\mathrm{d}t} \right)_{\text{growth}} = r_{\text{growth}} \cdot A \cdot \rho + \lambda, \tag{12}$$

but if $\rho \geq \rho_{\text{threshold}}$ the growth rate is not affected by the likelihood parameter. The likelihood threshold $\rho_{\text{threshold}}$ is used to distinguish between well protected and poorly protected coastal zones. This is important as we assume that the amount of asset growth that is removed due to reduced investment is not completely lost from all coastal zones, but might be invested into other, better protected, coastal zones, which is reflected by the parameter $\lambda$. We crudely assume that $50 \pm 30\%$ of investments are preferably made in the coastal zone, so that this part of the removed growth from badly protected coastal zones is actually moved to coastal zones with $\rho \geq \rho_{\text{threshold}}$, weighted by their relative assets fraction. If there are no such coastal zones, or if the model is used in global mode, i.e. with just one global coastal zone, the coast-specific part of removed growth is added back to the badly protected coastal zones, under the assumption that investment will then be made regardless of the risk. Even though this is a stylized implementation of human behaviour, it still represents a significant step forward from the typical assumption that there is no feedback from expected SLR exposure to asset growth in the coastal zone that is typically used in IAMs. Nevertheless, the reduction of asset growth in coastal zones with expected increasing flood risk is turned off in our comparison to CIAM, as this is a hypothetical behaviour that is not captured by CIAM.

2. SLR-driven storm surge damages: The increase of storm surge damages under rising sea levels is conceivably the main component of the overall costs of SLR in a scenario of no adaptation against SLR (Wong et al., 2022). FRISIA calculates the fraction of the original coastal asset distribution that is generally susceptible to storm surge damages, $F_{\text{sus}}^{\text{A}}$, as well as the respective fraction of assets that experiences storm surge damages in a given year, $F_{\text{exp}}^{\text{A}}$, as functions of the effective flood height $S$ that were fitted to CIAM data of annual storm surge exposure (Fig. 3). From the latter, the difference between the current and the initial exposure fraction is used to only account for storm surge damages that are driven by SLR. Furthermore, the asset fraction that is already removed from the coastal zone by forced or planned retreat, $F_{\text{rem}}^{\text{A}}$, is taken into account to calculate the actually exposed asset fraction in a given year (a feature that is not accounted for by CIAM). The overall annual storm surge damage is then

$$D_{\text{storm}} = A \cdot f_{\text{maxDamage}} \cdot (1 - R) \quad \cdot \left( \frac{F_{\text{sus}}^{\text{A}} - F_{\text{rem}}^{\text{A}}}{1 - F_{\text{rem}}^{\text{A}}} \right) \cdot \text{MAX} \left( 0, F_{\text{exp}}^{\text{A}} - F_{\text{exp,t=0}}^{\text{A}} \right), \tag{13}$$

where the flood damage resilience $R$ is a function of GDP per capita as in CIAM, accounting for the fact that wealthier coastal zones are more resilient to damages (Diaz, 2016). We further added the factor $f_{\text{maxDamage}} = 0.3 \pm 0.1$, which is the maximum fraction of exposed asset values that can be destroyed in a storm surge at zero resilience. This parameter is used to calibrate our results against CIAM output of storm surge damages.

While the standard assumption in most existing SLR impact models is that these damages do not impact the evolution of coastal assets, there is growing evidence for lasting reductions at least in real-estate values after major storm surge events (e.g. Fisher and Rutledge, 2021; Addoum et al., 2024; Holtermans et al., 2024); see Contat et al. (2024) for a recent review. Adding to this is the possibility that firms might not rebuilt or further invest into damaged company sites due to the potential of future

flooding, we introduce a very simple formulation in which only a fraction (0.75–1.0) of damaged assets is repaired. The actual reduction of coastal asset values then becomes

$$\left(\frac{\mathrm{d}A}{\mathrm{d}t}\right)_{\text{storm damage}} = -D_{\text{storm}} \cdot (1 - f_{\text{repair}}) \,. \tag{14}$$

In the reference setup without additional feedback, which we use to compare FRISIA to CIAM, we set the repaired fraction $f_{\text{repair}} = 1$, so that storm surge damages are not reducing coastal asset values. We further explore the consequences of setting $f_{\text{repair}} < 1$ in Sect. 5.1.

3. Planned retreat: The formulation of planned retreat of assets uses the expected effective flood height in 50 years to calculate the expected future susceptible asset fraction, $F_{\text{sus},50}^{\text{A}}$ using the same fitted function as for calculating $F_{\text{sus}}^{\text{A}}$. Furthermore, just as for the protection decision, we use a scale parameter $W_{\text{retreat}}$, which can be seen as the population's will to retreat under expected storm surge exposure. Assets can conduct planned retreat if they will generally be susceptible to storm surges in the future, as owners will try to reduce future damages from SLR. This is a strong form of retreat that is comparable to a retreat behind the 1 in 1000 year storm surge height in CIAM. The change to coastal assets because of planned retreat is

$$\left(\frac{\mathrm{d}A}{\mathrm{d}t}\right)_{\text{planned retreat}} = -W_{\text{retreat}} \cdot \frac{A}{\tau_{\text{retreat}}} \cdot \text{MAX}\left(0, \frac{F_{\text{sus},50}^{\text{A}} - F_{\text{rem}}^{\text{A}}}{1 - F_{\text{rem}}^{\text{A}}}\right), \tag{15}$$

where $\tau_{\text{retreat}} = 10\ (5-25)$ years is the proactive retreat time scale of coastal assets.

4. Forced retreat: this form of retreat occurs for assets that are becoming inundated because of sea level rise. We calculate the inundated asset fraction $F_{\text{ind}}^{\text{A}}$, similar to the susceptible and exposed fractions, via a logistic function fitted to DIVA data. The change of the coastal asset stock due to forced retreat then amounts to

$$\left(\frac{\mathrm{d}A}{\mathrm{d}t}\right)_{\text{forced retreat}} = -A \cdot \text{MAX}\left(0, \frac{F_{\text{ind}}^{\text{A}} - F_{\text{rem}}^{\text{A}}}{1 - F_{\text{rem}}^{\text{A}}}\right), \tag{16}$$

again taking into account the asset fraction that was already removed in previous time steps $F_{\text{rem}}^{\text{A}}$. Hence, when planned retreat was already undertaken, $F_{\text{rem}}^{\text{A}} > F_{\text{ind}}^{\text{A}}$, and the forced retreat is zero. In reality, asset values would be reduced earlier than modelled here via the depreciation of assets that are to become inundated. Therefore, the modelled reduction in the asset stock is lagging behind what would occur in a more complex model that incorporates this depreciation, but the final value over long time scales is ultimately the same.

### 3.2.5 Coastal population

FRISIA tracks a stock of coastal population that develops independently of the stock of coastal assets. Nevertheless, changes in coastal assets and population are driven by the same general processes, with just minor differences in parameterization. The coastal population stock is initialised with population data from each coastal segment, taken from the DIVA dataset (Vafeidis et al., 2008). Coastal population, $P$, then grows with a global or regional coastal growth rate, which is provided by the IAM, or prescribed externally when the model is run uncoupled:

$$\left(\frac{\mathrm{d}P}{\mathrm{d}t}\right)_{\text{growth}} = r_{\text{P}} \cdot P \tag{17}$$

Unlike in the case of assets, we do not assume a possible reduction in this growth rate under expected increased future flood exposure, and it should be noted that the population growth can become negative in the future.

The population fraction that is generally susceptible and the fraction that is annually exposed to storm surges are both calculated from a logistic function fitted to DIVA and CIAM data, using the same method as the respective asset fraction (Fig. 3). The number of people exposed to storm surges because of SLR, $P_{\text{exposed}}$, is calculated similarly to asset storm surge damages:

$$P_{\text{exposed}} = P \cdot \left( \frac{F_{\text{sus}}^{\text{P}} - F_{\text{rem}}^{\text{P}}}{1 - F_{\text{rem}}^{\text{P}}} \right) \cdot \text{MAX}\left( 0, F_{\text{exp}}^{\text{P}}(t) - F_{\text{exp}}^{\text{P}}(t=0) \right). \tag{18}$$

Again, we use the difference between the current and the initial exposure fraction of people to only track the SLR-driven number of exposed people. The change in the coastal population stock due to storm surges equals the number of fatalities in storm surges

$$\left( \frac{dP}{dt} \right)_{\text{storm loss}} = -f_{\text{fatality}} \cdot (1 - R) \cdot P_{\text{exposed}}. \tag{19}$$

The fatality fraction $f_{\text{fatality}} = 0.01 \, (0.005 - 0.02)$ of people exposed to storm surges is the same as in CIAM and $R$ is the same resilience as for the damage to coastal assets. While CIAM has the same fatality rate, flood fatalities do not feed back to the population density in CIAM.

Planned and forced population retreat are handled in exactly the same way as retreat of assets:

$$\left( \frac{dP}{dt} \right)_{\text{planned retreat}} = -W_{\text{retreat}} \cdot \frac{P}{\tau_{\text{retreat}}} \cdot \text{MAX}\left( 0, \frac{F_{\text{sus,50}}^{\text{P}} - F_{\text{rem}}^{\text{P}}}{1 - F_{\text{rem}}^{\text{P}}} \right), \tag{20}$$

$$\left( \frac{dP}{dt} \right)_{\text{forced retreat}} = -P \cdot \text{MAX}\left( 0, \frac{F_{\text{ind}}^{\text{P}} - F_{\text{rem}}^{\text{P}}}{1 - F_{\text{rem}}^{\text{P}}} \right). \tag{21}$$

This uses the same retreat time scale and scale parameter as for the retreat of assets.

## 4 Costs of sea level rise in FRISIAv1.0

### 4.1 Cost formulation

We define the costs and damages of sea level rise principally in the same way as Wong et al. (2022). However, we do not translate the loss of life into a cost, but rather compare the number of annual storm surge fatalities between the models. When FRISIA is coupled to an IAM the loss of life can feed into the integrated population model, and be counted as part of the loss of life due to climate impacts. Apart from that, there are four different types of costs:

1. The costs of protecting against SLR, which are defined as the sum of the costs to construct new flood protection, the costs to maintain the additional flood protection and the opportunity costs of the land lost for the new flood protection, which are defined in section 3.2.3.

$$\text{Cost}_{\text{protect}} = \text{Cost}_{\text{protect,build}} + \text{Cost}_{\text{protect,maintain}} + \text{Cost}_{\text{protect,land}} \tag{22}$$

2. The costs of storm surge damages $D_{\text{storm}}$, as defined in section 3.2.4 (Eq. 13).

3. The cost to relocate people and assets, $\text{Cost}_{\text{relocate}}$, which is based on the people and assets that retreat annually. The total costs consist of three sub-costs, and all our cost formulations are in line with assumptions made in CIAM, except that we added uncertainty ranges to better capture the parametric uncertainty. First, there is the cost to relocate people, which for people who are conducting a planned retreat is the average annual income per person and is $\gamma = 4 \pm 1$ times as much for people that are forced to retreat (CIAM applies a constant increase factor of 5, noting that this parameter, as well as the following ones, lack an empirical basis, see supplementary material in Diaz (2016). Lincke and Hinkel (2021, supporting information Tab. S4) reviews literature values ranging from 2.3 to 9.5). Second, there is the cost to demolish the abandoned immobile part of the assets, and these demolition costs are assumed to be $c_{\text{demolition}} = 0.05 \pm 0.025$ of the immobile part of asset values. Third, there is the cost to relocate the mobile part of assets. We assume the mobile fraction of coastal assets to be $f_{\text{mobile}} = 0.25 \pm 0.05$. The cost to relocate the mobile assets is assumed to be a fraction of the retreating asset values, $c_{\text{relocate,A}} = 0.1 \pm 0.05$.

$$\text{Cost}_{\text{relocate,P}} = \left[ \left( \frac{dP}{dt} \right)_{\text{planned retreat}} + \gamma \cdot \left( \frac{dP}{dt} \right)_{\text{forced retreat}} \right] \cdot \text{GDP}_{\text{perCapita}} \tag{23}$$

$$\text{Cost}_{\text{relocate,A}} = \left( \frac{dA}{dt} \right)_{\text{retreat}} \cdot f_{\text{mobile}} \cdot c_{\text{relocate,A}} \tag{24}$$

$$\text{Cost}_{\text{demolition}} = \left( \frac{dA}{dt} \right)_{\text{retreat}} \cdot (1 - f_{\text{mobile}}) \cdot c_{\text{demolition}} \tag{25}$$

$$\text{Cost}_{\text{relocate}} = \text{Cost}_{\text{relocate,P}} + \text{Cost}_{\text{relocate,A}} + \text{Cost}_{\text{demolition}} \tag{26}$$

4. The total cost of flooding is the sum of the value of the abandoned immobile part of coastal assets and the opportunity costs of the land that is lost to the sea. We assume that under planned retreat a large part of the immobile assets is depreciated at the time of retreat, so that only a small fraction of those assets, $f_{\text{remaining}} = 0.1 \pm 0.1$, contributes to the costs of flooding. Under forced retreat, the full immobile asset values are lost and contribute to this cost.

$$\text{Cost}_{\text{flooding,A}} = (1 - f_{\text{mobile}}) \cdot \left[ f_{\text{remaining}} \cdot \left( \frac{dA}{dt} \right)_{\text{planned retreat}} + \left( \frac{dA}{dt} \right)_{\text{forced retreat}} \right] \tag{27}$$

$$\text{Cost}_{\text{flooding,land}} = f_{\text{land}} \cdot lv \cdot \max \left( \text{Land}_{\text{inundated}}, \text{Land}_{\text{abandoned}} \right) \tag{28}$$

$$\tag{29}$$

A logistic function that is fitted to DIVA data is thereby used to calculate the area of the land that is inundated or abandoned in the case of retreat, as described in Sect. 3.2.2.

The costs of wetland losses are not included in FRISIA. We have left this scope for future development because the current generation of IAMs is unable to make use of this information. Furthermore, we note that, just like CIAM, our model does not capture the effect of SLR on economies that interact with the coastal environment, like fisheries, tourism or international

trade. There is insufficient information available to appropriately calibrate and represent the indirect impacts of SLR on coastal industries. These costs are only partly represented via their relationship to coastal GDP and hence asset values, so that the calculated costs in FRISIAv1.0 are potentially underestimated.

## 4.2 CIAM comparison

For the evaluation of the model, we set up FRISIA as close as possible to the CIAM setup presented in Wong et al. (2022). The simulations start in the year 2010 and integrate with a time step of one year until the year 2150. The population and asset stocks are initialised from SSP and DIVA data as described in the sections above. We note that the data assume a growth rate of zero for assets and population after 2100, which is the same as in Wong et al. (2022), because the long-term socio-economic evolution is very uncertain. This continuation with constant population and GDP leads to a visible kink in the time series, and the form of data continuation needs to be considered when interpreting the results. Furthermore, it should be noted that CIAM integrates with a time step of ten years and costs are partly spread over the 40-50 year adaptation periods (Wong et al., 2022). This makes a direct comparison to FRISIA ambiguous, as FRISIA calculates with a one year time step, and costs are calculated for each year individually. Nevertheless, a more general comparison that focuses on the temporal evolution and the broad range of individual costs is still useful. We note that CIAM and FRISIA will produce different results not only due to differences in level of aggregation but also due to structural changes introduced in FRISIA like the use of the small time step and differently spreading costs over time. Lastly, we run the model for the same SSP scenarios and use the same prescribed (non-optimized) adaptation strategies as presented in Wong et al. (2022). Namely, we run the SSP1-2.6, SSP2-4.5 and SSP5-8.5 scenarios (Riahi et al., 2017) and apply three different adaptation strategies: *No Adaptation*, *Protect* and *Retreat*.

Figure 4 shows the costs for the *No Adaptation* strategy, while Fig. 5 displays the costs of the *Protect* and *Retreat* strategies. For each strategy, only the relevant non-zero costs are plotted. It should be noted that, despite having the same adaptation strategies in both models, there are some inherent differences between FRISIA and the CIAM setup in Wong et al. (2022). First, Wong et al. (2022) initialise the model with the respective adaptation strategy in the first time step. This means that for the *No Adaptation* strategy, their model assumes that there are no dikes or other forms of flood protection, or that for a *Retreat* strategy, assets and population have retreated above the height of a storm surge with a specific return period. In contrast to that, we do not recalculate the initial retreat or protect level, but directly use the DIVA data for model initialisation, including an initial flood protection. This means FRISIA by default includes an initial flood protection even when applying the *No Adaptation* or *Retreat* strategies, which stands in contrast to what is done in Wong et al. (2022). Therefore, for the comparison in this section, we add another setup of FRISIA, in which there is no initial flood protection. The simulations using this setup are represented by the dotted lines in Figs. 4 and 5. Structurally, the main difference to the standard FRISIA setup is that the model now responds differently to increasing sea levels, represented by the fitted functions for how the asset's and population's annual exposure and susceptibility fractions behave as a function of the effective flood height. In order to then calculate just those costs that are induced by SLR, we subtract the initial storm surge exposure from the current exposure in the storm surge damage calculation. Second, CIAM allows for decisions to retreat or protect against storm surge heights with different return periods of 1, 10, 100 or 1000 years. We do not incorporate that level of detail, but rather include the parameters $W_{\text{protect}}$ and

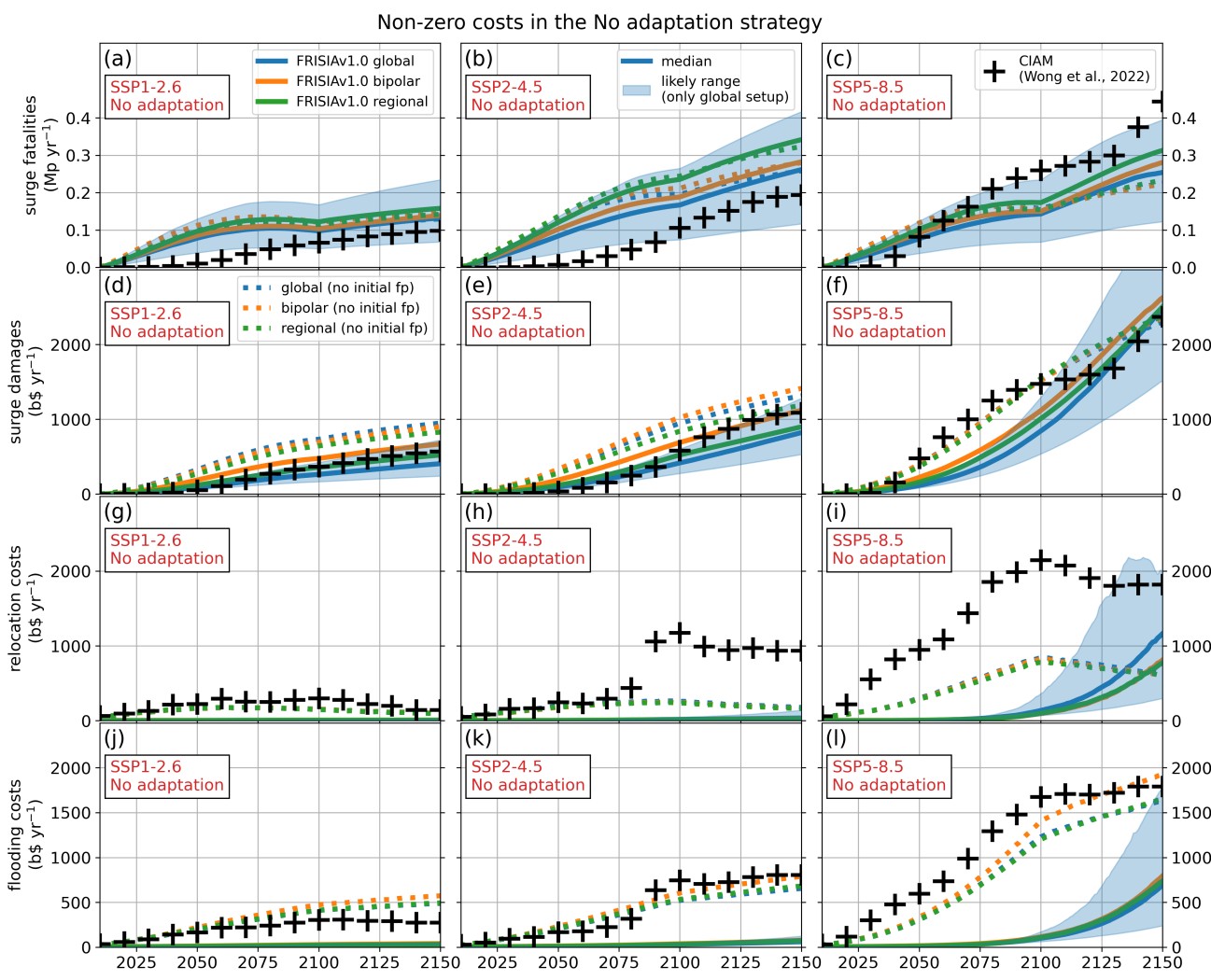

**Figure 4.** The global costs of sea level rise in the *No Adaptation* strategy. Data from CIAM (Wong et al., 2022) are compared to output from FRISIA for different levels of aggregation. Shown are results for the scenarios SSP1-2.6 (left), SSP2-4.5 (middle) and SSP5-8.5 (right). The solid lines represent model runs with an initial flood protection height as taken from the DIVA dataset, whereas the dotted lines are the same simulations with an initial flood protection height of zero, as in CIAM. The shaded area represents the $17^{th} - 83^{rd}$ percentile range from the global FRISIA setup with initial flood protection.

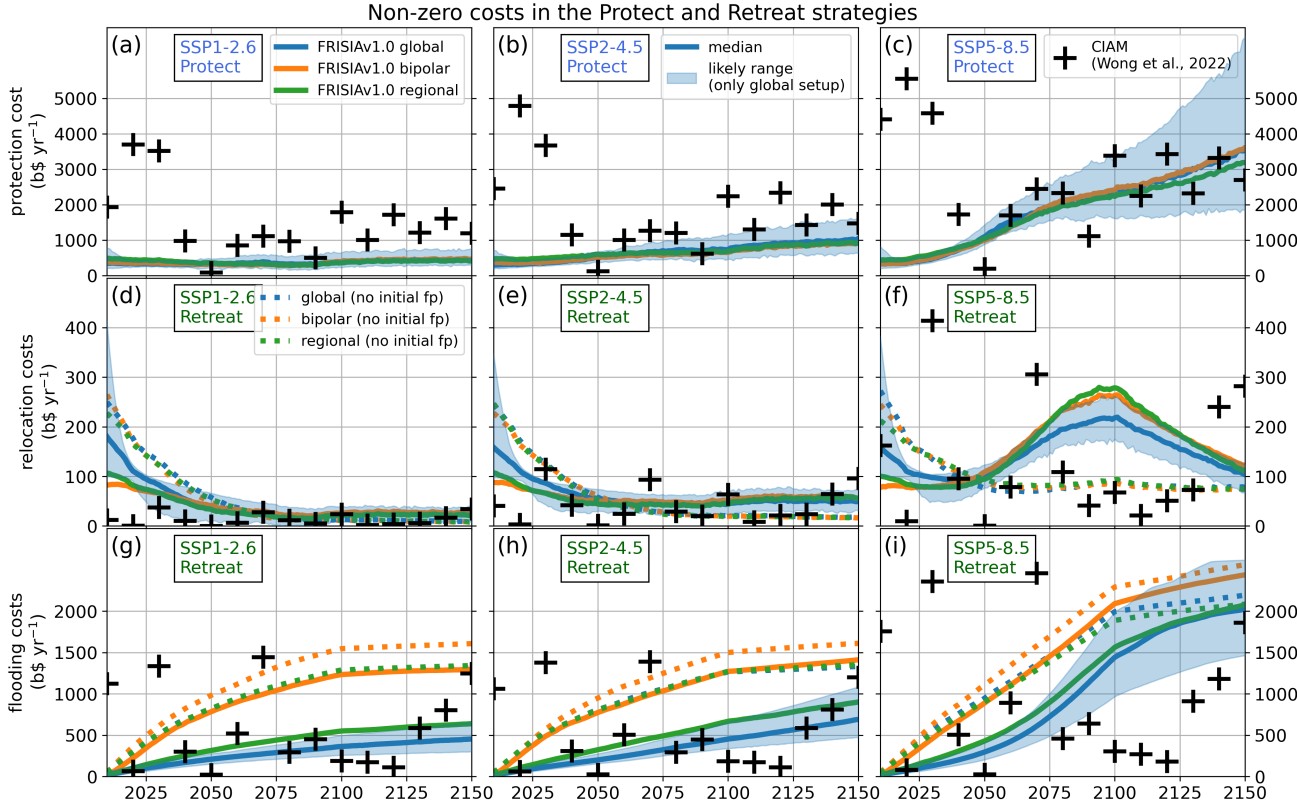

**Figure 5.** The global costs of sea level rise in the *Protect* (a)–(c) and *Retreat* (d)–(i) adaptation strategies. All figure settings are as in 4. Only the non-zero costs are shown, so for example in the "Protect" scenario, there are no storm surge damages or fatalities.

$W_{\text{retreat}}$, which can be varied between zero and one. A value of one thereby indicates a desired perfect protection against any SLR-driven damages in the *Protect* case and a retreat to the 1000 year maximum storm surge height in the *Retreat* case. Hence, the results depicted in Fig. 5 show data from the maximum possible *Retreat* or *Protect* strategies for both CIAM and FRISIA. For each adaptation strategy, scenario and ensemble member, the FRISIA model was run in three setups with different levels

of aggregation: a global model, a bipolar model, and a regional model (see Sect. 3.2.1 and Table A2) in the appendix.

In the *No Adaptation* strategy, global damages in FRISIA are primarily driven by increasing storm surge damages to population and assets, while relocation and flooding (inundation) costs become relevant only in the 22$^{\text{nd}}$ century. In contrast to that, all costs are of a similar importance in CIAM, mostly because there is no initial flood protection at all, so that land becomes inundated and people and assets have to be relocated much earlier. This fits well with our additional experiments without any

initial flood protection in FRISIA, which are depicted as the dotted lines in Fig. 4. In these runs, FRISIA relocation and flooding costs are very similar to those in CIAM, but generally a bit lower in the later part of the simulation. Especially, relocation costs are much lower in FRISIA in SSP5-8.5 and after 2090 in SSP2-4.5, but there is no single driver that can explain those differences between the models. One aspect is that the SLR model used in Wong et al. (2022) produces GMSL projections that

are higher than the ones in FRISIA by some 10 cm in 2100. Another might be the specific way that costs are aggregated over time in CIAM. We note that the projections from FRISIA draw a more consistent picture in terms of the temporal evolution of costs. After all, the setup with no initial flood protection is a counterfactual scenario.

Similarly, storm surge damages are higher in the early part of the simulation without initial flood protection. This is because in this simulation all parts of the coastal zone will experience early damages from SLR. However, the damages then grow slower, because the additional effect that the initial flood protection can be breached in the future is missing. For the remaining part of the discussion, we stick to the results of the FRISIA setups that do include the initial flood protection, under the assumption that existing dikes will be maintained even in a *No Adaptation* strategy.

The time series of SLR-driven storm surge fatalities shows a visible kink and faster increase after 2100, while storm surge damages to assets do not show this behaviour. The reason for the kink is the continuation of input data with constant values as described above. But what is the driver of the change in behaviour? The global coastal population is decreasing before 2100, while GDP increases. This leads to a decreasing number of people being potentially susceptible, and, at the same time, an increase of the resilience to damages due to a higher GDP per capita. Both trends counteract the increase of storm surge fatalities due to SLR. After 2100, these trends stop so that the remaining driver is the increase of storm surge fatalities from SLR. The difference for assets is that before 2100 damages are already accelerating, driven by SLR and the increase of asset values with GDP. After 2100 the continuation with constant values slows down the increase of asset values, but also storm surge resilience ceases to increase. Hence, there is no visible change in the evolution of storm surge damage to assets, despite a substantial change in the underlying socio-economic dynamics.

The costs of protecting against SLR are generally the same order of magnitude in FRISIA and CIAM (Fig. 5, top row). The main difference is that there are no jumps in protection costs in FRISIA, as the model simulates more continuous costs than the ten year time step and the adaptation periods in CIAM would allow (the CIAM version presented in Wong et al. (2022) assumes adaptation periods of 40-50 years, starting in 2010, 2050 and 2100, and the flood protection costs are calculated at the start of each adaptation period, assuming that knowledge about future SLR is limited to the end of the adaptation period). The global protection costs are the only relevant costs in the *Protect* strategy with FRISIA, as the costs of losing wetlands are not calculated in the model. Generally, the total annual protection costs are smaller than the sum of all costs in a *No Adaptation* strategy at the end of the simulation period, independent of the aggregation level, but flood protection requires investment that is higher than the costs in *No Adaptation* during the earliest part of the simulation period.

In the *Retreat* strategy, FRISIA calculates high initial relocation costs because, in the chosen case of maximum retreat, all coastal assets and people retreat to a height where they are not susceptible to storm surges at all. While this is generally the same assumption as in the respective CIAM simulation depicted in Fig. 5, the early relocation costs in CIAM are much smaller, because CIAM assumes an initial state that already follows the chosen adaptation strategy. In FRISIA this adaptation has to happen in the simulation itself, hence there is an initial spike in relocation costs. After this transient behaviour, FRISIA and CIAM calculate similar relocation costs, but the FRISIA projections follow a much smoother curve, because of the reasons discussed above.

Interestingly, FRISIA simulates a peak of relocation costs under the *Retreat* strategy in 2100, but only for SSP5-8.5 (Fig. 5 (f)). Relocation costs are defined as the costs to move mobile assets and people, and the costs to demolish abandoned immobile assets. These costs are proportional to GDP (assets) and GDP per capita (people), both of which stop to increase after 2100, as a consequence of the chosen continuation method of SSP data beyond that year (compare Fig. A7). This explains one part of why relocation costs stop growing after 2100, and also why the increase of flooding costs slows down in that year (Fig. 5 (i)). However, the question remains why relocation costs increase in SSP5-8.5 between approximately 2030 and 2100, while they remain stable or decrease in the other scenarios. One driver for this is that generally GDP increases a lot faster in SSP5-8.5, where it ends up being almost a factor of two higher in 2100 than in the other scenarios before considering retreat. It also keeps accelerating, while GDP grows more linearly in SSP2-4.5 and growth even decelerates in SSP1-2.6 (this is also a driver of the much faster increase of surge damages in SSP5-8.5 under no adaptation, Fig. 4 (d-f)). The final component necessary to describe the peak shape in relocation costs is the much stronger SLR in SSP5-8.5, which drives the same shape, albeit in weaker form, also in relative terms for asset retreat (not shown). The insight here is that under a strong (expected) SLR many assets that are initially safe because of the currently existing flood protection will become susceptible to storm surge damages in the course of the 21$^{st}$ century and hence retreat. This happens in a relatively short amount of time in SSP5-8.5, while it is more spread in the other scenarios due to slower SLR. The peak shape is therefore also a consequence of including the initial flood protection from the DIVA dataset even in our *Retreat* strategy simulations, which has not been included in CIAM. Correspondingly, the counter-factual scenario of no initial flood protection (dotted lines), does not show the peak shape, as here assets and people retreat already in the first years of the simulation (at lower costs, but much stronger in relative terms).

The final form of SLR costs, the costs of flooding in the *Retreat* strategy, are also similar between FRISIA and CIAM, but have less variation in FRISIA. Here, there is a noticeable difference between the three FRISIA setups, where the bipolar setup leads to higher flooding costs, especially in the early years, when SLR is relatively small. The difference between the model versions thereby comes from aggregation errors when converting retreating assets into abandoned land, and these errors are the lowest in the bipolar setup. This issue only affects the opportunity costs of the abandoned land, which do not provide feedback to any other part of FRISIA, and which are also not likely to be useful output in most IAMs. Therefore, we do not act on this dependence. Fixing the problem would require a substantial increase of process detail, and in the current version of the model we put a higher value on dynamic complexity and maintaining model simplicity.

Overall, the presented simulations show that the FRISIA model calculates estimates for the costs of sea level rise that are in line with the projections of the more complex CIAM model (Wong et al., 2022). FRISIA can produce SLR damages and adaptation costs in scenarios of no adaptation, protection or retreat from the coast, and the relevant FRISIA output is largely independent of the level of aggregation.

## 5 Incorporating coastal feedbacks

The previous section has shown that the no-feedback version of FRISIAv1.0 can approximate the calculated global costs of SLR impacts and adaptation in CIAM (Wong et al., 2022), when given the same input. In this section, we make use of the

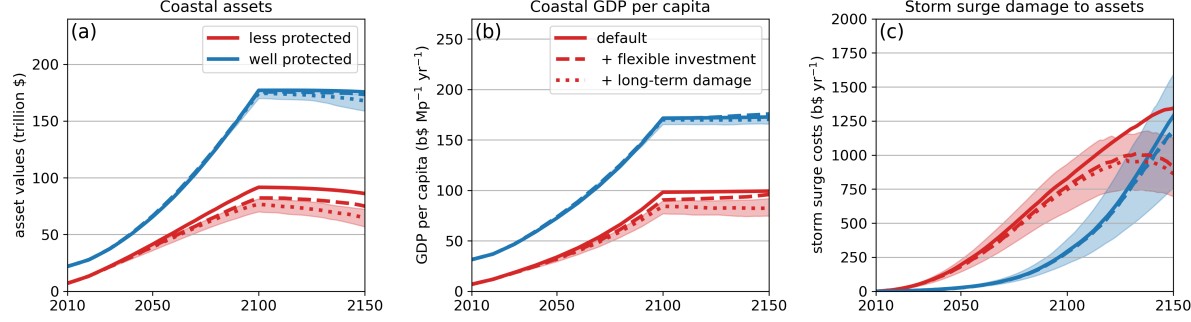

**Figure 6.** The effect of asset feedbacks on the evolution of coastal asset values (a), GDP per capita (b) and annual storm surge damages (c) for the *No Adaptation* strategy in the bipolar setup. Shown are the median values of the no-feedback case presented in section 4, and two cases with additional feedbacks as described in the text. The shaded areas represent the 17th - 83rd percentile range of the final setup only, as showing all uncertainty ranges would obscure the figure. All input data are from SSP5-8.5. We point out again that the almost constant evolution of coastal assets and GDP per capita after 2100 is because the reference GDP and population data from the SSP scenario are continued with constant values after that year.

non-optimising, flexible model structure of FRISIA, and explore the effect of incorporating additional feedback that could influence the evolution of coastal assets or population, and hence the costs of SLR. For this, we use the bipolar model setup of FRISIA, because its unequal separation of assets and population can lead to heterogeneous dynamics, while still maintaining a high level of aggregation. This makes it more illustrative than the regional setup of FRISIA, in line with the focus of this study
being to showcase the capabilities of FRISIA and to understand the dynamics of the coastal zone under rising sea levels. For the same reason, we use the SSP5-8.5 scenario in this section, as it is the most extreme case of SLR.

## 5.1    Reduced future investments and reparation

CIAM and the no-feedback version of FRISIAv1.0 assume that storm surge damages have no long-term effect on the evolution of coastal assets and are immediately repaired. Furthermore, the structure in CIAM represents the assumption that assets in the
coastal zone grow independently of the expected SLR and storm surge exposure, based on the regional GDP growth taken from SSP input data. Here, we explore what happens in the case that both of these effects lead to a reduction in asset values or their growth rate under the *No Adaptation* strategy. The underlying philosophy of the new feedbacks has been laid out in Sect. 3.2.4. To implement these effects, the evolution of coastal GDP is made dependent on the evolution of coastal assets in all scenarios presented from this point forward. A relative reduction in coastal asset values compared to the coastal population will now
reduce the average GDP per capita in the coastal zone, and thereby reduce the coastal resilience to storm surge damages in the future. From this point forwards, all scenarios also include the effect that asset loss through inundation reduces the fraction of assets annually exposed to storm surges.

If a coastal zone expects an increase in exposure to storm surge damages in the future, that is $S_{50} > 0$ in FRISIA, it is conceivable that there will be less investment into the respective coastal zone. FRISIA makes the simplifying, but reasonable, assumption that asset growth is a form of investment, therefore, incorporating this behavioural feedback translates into a reduction in the annual growth of asset values, as reflected in Eq. 11 and 12. Furthermore, within FRISIA we assume that some of the growth reduction is actually moved to coastal zones that retain a low exposure to storm surge damages. The effect of incorporating this feedback is reflected by the difference between the solid and the dashed curves in Fig. 6 (additional time series of variables can be found in the appendix Fig. A9). In the bipolar aggregation setup of FRISIA, by definition, all assets in the less protected coastal zones, but none in the well protected coastal zones, are initially susceptible to storm surge damages. Hence, this mainly affects less protected coastal zones, in which coastal asset values are reduced by 11.2 (6.4 - 15.6)% in 2150 compared to the baseline version without this feedback. The reduction in asset values then leads to smaller storm surge damages in the future, as there are less assets in the future. Because of this feedback, FRISIA simulates a peak in the SLR-driven storm surge damages in the first half of the 22$^{nd}$ century in less protected coastal zones. At the same time, a small part of the asset growth is moved from the less protected to the well protected coastal zone, slightly increasing the coastal asset values there. Therefore, activating the investment feedback has almost no effect on well protected coastal zones during the simulated period.

A peak of SLR damages in the first half of the 22$^{nd}$ century has also been simulated in the model of Desmet et al. (2021). They use a dynamic spatial economic model that allows capital and people to move to other locations, in search of the most economically efficient adaptation. Although they use a much more sophisticated economic model, this is in principle a similar feedback as simulated with our approximated simple reduction in the growth of coastal assets, resulting in the qualitative peak and decline shape of SLR damages. These results underline the importance of including migration into studies of SLR impacts.

Similar to a potential reduction in the growth of coastal assets, it is also possible that assets that have been damaged in a storm surge are not repaired. As a form of adaptation, some of the damaged assets might be rebuilt away from the coast, or not repaired at all. In this case, storm surge damages would actually reduce coastal asset values, effectively reducing coastal GDP and hence the resilience to storm surge damages in the future. The dotted line in Fig. 6 shows what happens if this feedback is activated in addition to the feedback described above. Again, the less protected coastal zones are the areas most affected. In these zones, coastal asset values are reduced by an additional 11.5 (4.3-21.7)% at the end of the simulation period, under a *No Adaptation* strategy and compared to the previous discussed case, where only the investment feedback is included. In the well protected coastal zone, the resulting coastal asset value reduction is just about 2.5 (1.0-6.1)%, because the SLR-driven storm surge damage is generally smaller in the early simulation period. The asset value reduction also leads to a reduction in future storm surge damages in the less protected coastal zone, as the amount of assets that can be damaged is lower. However, this reduction is smaller than what could be expected from the corresponding reduction in asset values, and there is no reduction in storm surge damages in the well protected coastal zone at all. This is a consequence of the reduced GDP per capita in the coastal zone, which leads to a reduction in resilience against storm surge damages.

Both of the above discussed feedbacks lead to a reduction in coastal assets and GDP, which, on the one hand, reduces the future storm surge damage potential, but, on the other hand, also reduces the resilience of the coastal zone against storm surge damages. We highlight that there is also a process that leads to the opposite effect, which is migration towards the coast or

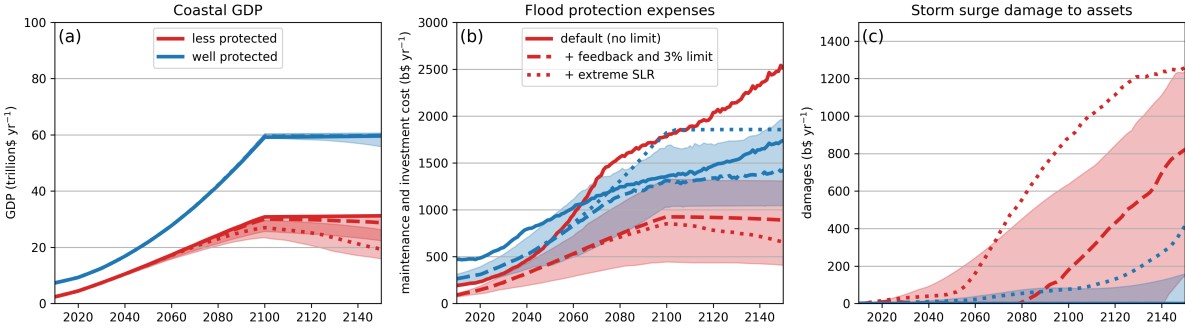

**Figure 7.** The effects of limiting flood protection spending and extreme SLR on the evolution of coastal GDP (a), flood protection expenses (b) and annual storm surge damages (c) for the *Protect* strategy in the bipolar setup. Shown are the median lines of three scenarios: 1. the no-feedback case presented in section 4 (solid lines). 2. A scenario with the additional feedbacks from Fig. 6 and a flood protection spending limit as a percentage of coastal GDP (dashed) 3. A scenario of extreme SLR with settings as in the previous scenario (dotted). All input data are from SSP5-8.5. The shaded areas represent the $17^{th}$ - $83^{rd}$ percentile range of scenario 2, as showing all uncertainty ranges would obscure the figure.

further urbanisation of coastal cities. Including coastal migration would increase coastal asset values and population, leading to potentially larger damages in the future. But whether coastal migration increases or decreases coastal GDP per capita will depend on the relative ratio of the growth rates. This effect is not fully included yet, as the growth rates of coastal assets and population are aggregated from country-level data, which do not account for relative changes in the asset and population distributions within a country. Incorporating coastal migration is therefore an interesting potential future application of FRISIA.

### 5.2 Failing to protect

So far, the *Protect* strategy in FRISIA allowed for unlimited investment into flood protection, neglecting the possibility that there is not enough money available. Here, we add a simple limit that annual flood protection spending is limited to 1-5% of the coastal GDP ($f_{\text{invest}}$ in Eq. 9). For completeness, we also include the previously introduced investment and storm surge damage feedbacks.

The difference between the solid and the dashed lines in Fig. 7 represents the impact of the flood protection investment limit. Now, in the early years, both, well and less protected coastal zones cannot invest the desired amount of money into flood protection. However, this does not lead to the emergence of SLR-driven storm surge damages in well protected coastal zones, as even the reduced investment leads to sufficient increases in the flood protection height that are ahead of increases in sea level. This is not the case for the less protected coastal zone, so that SLR-driven storm surge damages emerge around the year 2070. This then leads to a relative reduction in coastal asset values and GDP, limiting the available money for future flood protection investments even further. This process represents a positive feedback loop, in which less protected coastal zones do

not have enough money for full protection, leading to future SLR-driven damages that reduce the available amount of money even further.

So far, we looked at the median response and the likely ranges, driven by uncertainty in future warming and SLR. But how will the above discussed scenario of limited flood protection investment look in the worst case of maximum SLR and including the uncertain but high impact behaviour? For this, we set all contributions to GMSL rise to the maximum parameter values

assumed in FRISIA (see Sect. 3.1), leading to future GMSL rise close to the uppermost end of the IPCC projections (Fig. 1). The dotted line in Fig. 7 shows the impact of this extreme SLR scenario in the median scenario (the remaining uncertainty range, not shown here, comes from the uncertainty in temperature projections). In this scenario, even well protected coastal zones fail to fully protect against future SLR damages, and flood protection expenses continuously stay at the maximum amount of money that is available. In the less protected coastal zones, SLR-driven storm surge damages are never zero and increase

rapidly after 2060, despite continuous maximum investment in flood protection. Because of the large damages and the feedback discussed in the previous section, coastal GDP in the less protected coastal zones is only half of what it would be at the end of the simulation period in the model without investment limits. In the most extreme cases, where warming is also above the projected average, the limit on flood protection spending fails to even cover the maintenance costs in less protected coastal zones after around 2130, so that there will be no more investment in flood protection at all from that point on.

These simulations show that the straightforward assumption that there is a limit on the availability of money for building flood protection can substantially alter expected SLR impacts. It should be acknowledged that a regional or national government may provide additional funding to protect the local population or secure its access to the sea. But also this investment is limited. On the other hand, we did not include the potential that dikes fail, apart from the constant annual maintenance costs, or that maintenance costs rise as the sea level increases relative to the dike height. Another interesting aspect that becomes more

relevant in the context of integrated assessment modelling is the effect of using government spending for flood protection, which then cannot be used for other purposes. All of these are research questions that can relatively easily be addressed with FRISIA.

## 6   Incorporating dynamic SLR feedbacks in an IAM

Since FRISIA was developed mainly for the use in IAMs, we present an example of how the coupling to an IAM could be

done, and we explore the resulting socio-economic consequences. We use the newly developed Feedback-based knowledge Repository for Integrated Assessments (FRIDA) version 2.1 (Schoenberg et al., 2025), which is an IAM that follows the same system dynamic principles as our model. In fact, FRISIA is developed within the same project that currently develops FRIDA. As both models follow a similar modelling philosophy, the integration of FRISIA into FRIDA is straightforward. Nevertheless, the same connections should in principle be made in any other IAM, too.

We integrate the bipolar setup of FRISIA into FRIDA, as while FRIDA is a global model, this framework allows for heterogeneous coastal dynamics. FRIDA provides the global mean temperature anomaly and ocean heat content changes which are needed to calculate GMSL rise. In addition, the population-driven land water storage component of FRISIA is replaced. Here,

GMSL rise from land water storage changes is modelled directly using the groundwater anomaly from water extraction and the amount of water storage in hydropower dams, which are both provided by FRIDA. The latter uses a simple linear relationship between the installed hydropower energy capacity and the corresponding negative contribution to GMSL.

The impact and adaptation module of FRISIA also uses the current global mean temperature anomaly and $CO_2$ emissions to calculate the expected GMSL rise in 50 years. Furthermore, the global population growth rates and the growth rates of net bank assets are used for the evolution of coastal population and assets respectively. The use of these global variables implies that the FRISIA version coupled to the IAM FRIDA has no representation of coastal migration or urbanisation at all. Lastly, the coastal assets and population variables are initialised in the same way as discussed above, with the difference that the integration of the simulation of FRIDA starts in 1980, so that the initial values of our model are scaled down correspondingly.

In the following, we list all the ways in which FRISIA output is coupled back to the IAM. The implementation of these impact streams is described in more detail in Wells et al. (2025). These feedbacks represent a "tight coupling" between sectoral components in multisectoral modeling that are often neglected (Srikrishnan et al., 2022).

1. SLR-driven storm surge damages and assets lost during retreat or by inundation are assumed to cause a failure of safe loans in the financial module of FRIDA. In this module, banks issue loans. Issued loans form bank assets. A share of loans (unknown to the bank) are risky and will default. Loan failure causes the banks to tighten lending standards, slowing down growth, as well as leading to the firing of workers. Hence, as SLR-driven storm surge damages cause an increase in loan failures, this means they will eventually reduce global economic growth and hence global income, with all the corresponding subsequent impacts.

2. SLR-driven storm surge fatalities are translated into an additional global death rate, which is the same for all age groups.

3. The number of people affected by storm surges annually is translated into a reduction of global productivity. This follows the assumption made in the FUND model (Tol, 2007; Waldhoff et al., 2014), which is that people affected by storm surges will not work for two weeks. Accordingly, we reduce productivity of flooded workers by 0.96, but add an uncertainty range for this parameter (0.9-0.99).

4. The costs of relocating assets and people are added to global owner consumption in FRIDA, as it is the owners of assets that have to pay for the relocation. This affects the global circular flow of money, because there is then an additional transfer of money from owners bank accounts to firms' checking accounts.

5. The net amount of money spent on flood protection, that is, annual investment into flood protection and the costs of maintaining the additional part of the flood protection, is assumed to be planned government expenditure. This money is transferred to firms checking accounts, thereby adding to GDP, but not representing productive investment.

6. The exposure of the coastal population to SLR-driven storm surges increases the perception of climate risks, which - in FRIDAv2.1 - impacts the total demand for food and the demand for animal products. This follows the assumption that people can adapt their behaviour in response to experiencing the consequences of climate change. The developed model is described further in Rajah et al. (2025).

We run an uncertainty analysis of the FRIDA model, where all parameters of FRIDA and FRISIA are varied using the Sobol sequence method (Sobol, 1967), building an ensemble of model simulations with a size of 20,000 samples. The ranges for the parameters within FRISIA are thereby set as described in this manuscript. The result is a wide range of economic outcomes, which in the median is comparable to the SSP2-4.5 scenario, but with lower economic growth (see Fig. A10 in the appendix).

We run the full ensemble several times for the cases of no SLR impacts, all SLR impacts (*No Adaptation*), all SLR impacts (*Protect*) and each above-mentioned impact stream coupled individually (*No Adaptation*), always using the same sampling points. Hence, we get 20,000 sets of comparable realisations. From this ensemble, we draw all the realisations that lead to approximately 0.77 (0.75-0.79) m of GMSL rise by 2100 (the median value of IPCC AR6 projections from SSP5-8.5), leaving us with 1368 samples. We choose the SSP5-8.5 GMSL rise because for the analysis in this section we want a substantial increase of GMSL, which is still reached by enough samples to adequately cover the spectrum of possible socio-economic developments in FRIDA. This procedure allows us to estimate the effects of a well defined SLR scenario on a wide range of possible socio-economic futures. In Fig. 8 we show the global socio-economic effects of coupling the SLR impacts to FRIDA by presenting relative changes in global GDP, total inflation since 1980, annual deaths and government debt to GDP ratio, compared to the no impact simulation, in which all SLR impacts were turned off.

The above-described coupling of SLR costs from FRISIA to FRIDA leads to a reduction of global GDP by $1.5 - 6.2$ % (17th - 83rd percentile ranges) in the *No Adaptation* strategy by 2100. The most dominant effect by far is the SLR-driven defaulting of safe loans, causing more than 90 % of the overall reduction in GDP. All other impact channels are minor compared to that, with the SLR-driven reduction in worker productivity being the second largest effect, causing a reduction of global GDP by only $0.27 - 0.33$ %. The scenario in which only the SLR-driven effect on climate risk perception is coupled has a slight positive impact on GDP. Here, the perception of risk leads to a more sustainable consumption behaviour that lowers land-use $CO_2$ emissions, thereby reducing future temperature-driven damages (Rajah et al., 2025).

Apart from the effect on GDP, SLR impacts in the *No Adaptation* strategy also lead to a measurable increase in total inflation since 1980 of $0.9 - 4.2$ %, which is again mostly driven by the SLR impact on loan failures. Furthermore, in the *No Adaptation* case there is an increase of global annual deaths of $0.6 - 1.8$ % in 2100. Interestingly, only around a quarter of that is driven by the SLR-driven flood fatalities directly. The majority of the impact on global annual deaths again comes from the SLR effect on loan failures. This is because the global death rate in FRIDA is dependent on GDP. Hence, the largest demographic effect of SLR comes from the population becoming poorer, leading to a reduced life expectancy in our setup.

The *Protect* strategy shows similar initial reductions in global GDP as the *No Adaptation* strategy between 2010 and 2030. After that, global GDP shows a much less steep decline and arrives at a reduction of $0.5 - 3.2$ % compared to the run without SLR impacts. The global debt to GDP ratio does not increase in the *Protect* strategy relative to the reference case, despite continuously increasing protection spending. This is because the protection costs are assumed to be planned expenditure, redirected from other public expenditure. The *Protect* strategy still leads to some increase in total inflation compared to the setup without SLR impacts, as the total protection is limited (Sect. 5.2), and there are still SLR-driven asset damages.

Overall, the here estimated effect of SLR is within the range of what previous studies have suggested using other models (Hinkel et al., 2014; Brown et al., 2021; Cortés Arbués et al., 2024). Moreover, our example of coupling FRISIA to an IAM

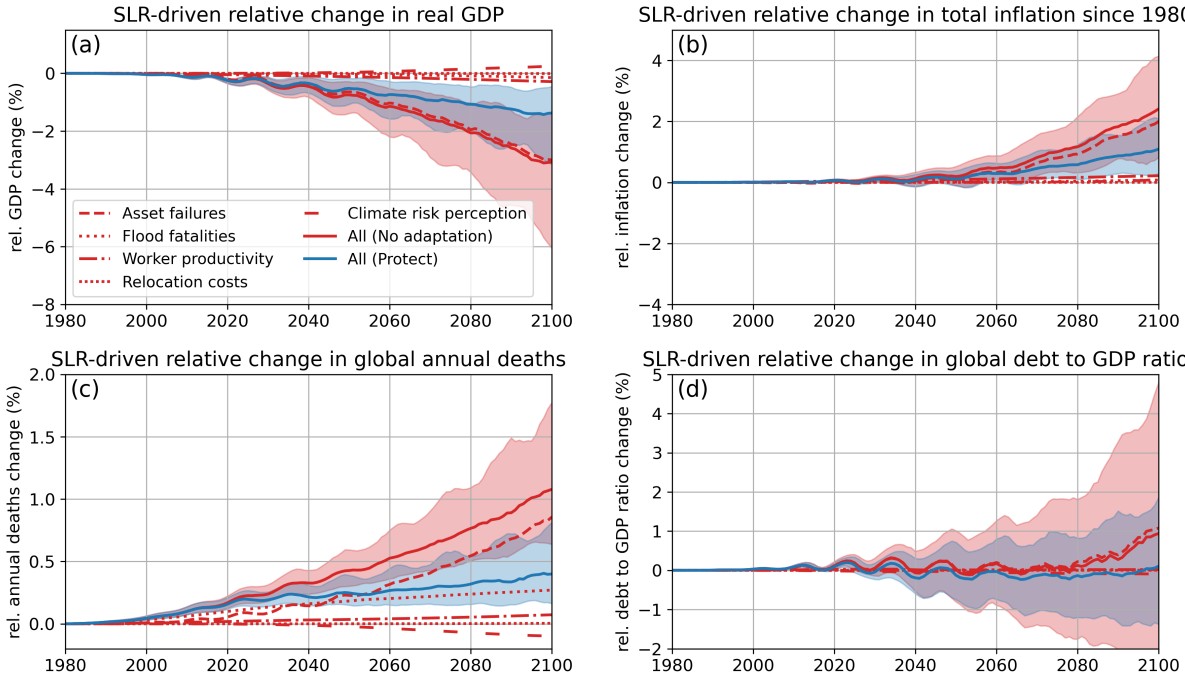

**Figure 8.** The global socio-economic effects of a SLR scenario that reaches approximately 0.77 (0.75-0.79) m of GMSL rise by 2100, as simulated within FRIDAv2.1, using SLR costs calculated by FRISIAv1.0. Shown are relative changes compared to a reference simulation without coupling of SLR impacts or adaptation costs in the following quantities: real GDP (a), total inflation since integration start in 1980 (b), global annual deaths (c), and global debt to GDP ratio (d).

highlights the diverse consequences that SLR can have on the economy and population. We end this section by noting that the calculated effects of SLR on the global economy and population are only a part of the full picture, as the socio-economic consequences in the coastal zone itself will be immense.

# 7 Discussion

## 7.1 Model uncertainty

The FRISIA model uses an ensemble of GMST projections from the FaIR model (Leach et al., 2021) as input and provides uncertainty ranges for parameters in the formulation of the individual SLR components. Hence, it fully covers the uncertainty in projections of GMSL to the degree that it is represented in the IPCC's sixth assessment report (Fox-Kemper et al., 2021), as we have shown here in Fig. 1 and in Figs. A1–A5. Nevertheless, uncertainty in the impacts of SLR is also to a large degree coming from uncertainty in the socio-economic evolution of the coastal zone. In Sect. 4 and 5, we have used country-level SSP data of GDP and population to project the evolution of coastal assets and population into the future, and the only accounting

for uncertainty in this regard was the use of different SSP scenarios in the uncoupled setup. There are several processes and aspects of uncertainty that this approach fails to cover: The socio-economic evolution may be more heterogenous between regions, e.g. not following the SSP2 or SSP5 scenario everywhere. Even within countries, there are substantial differences in population and GDP growth. Migration towards or away from the coast – even without considering retreat in response to SLR – will impact the evolution of coastal GDP and population stocks (Desmet et al., 2021), and the ratio of assets to GDP might change over time, impacting the estimated damages. Finally, the SSP scenarios might not cover the full spread of the potential future socio-economic evolution, especially on longer time scales. This final aspect can be accounted for by coupling FRISIA directly to an IAM and running a large uncertainty ensemble with the IAM, as we did in Sect. 6. Nevertheless, the other aspects mean that the true degree of uncertainty in SLR impacts is underestimated by our approach. However, this is the case for all studies that use SSP data, and many other SLR studies suffer from the same uncertainties (Hinkel et al., 2014; Lincke and Hinkel, 2018; Parrado et al., 2020; Lincke and Hinkel, 2021; Brown et al., 2021; Bachner et al., 2022; Magnan et al., 2022; Wong et al., 2022).

Apart from uncertainty in SLR and the socio-economic evolution of the coastal zone, there are also the parametric and structural uncertainties of the impacts and adaptation model of FRISIA. We have approached the former by assigning generous ranges instead of single values to essentially all parameters in the impacts and adaptation model of FRISIA for which there is uncertainty, excluding only those that represent aggregated information from the input models DIVA and CIAM. Model results are then always inferred from an ensemble of runs that randomly vary these parameters independently and interpreted in terms of median values or likely ranges. We believe that this approach covers the parametric uncertainty of FRISIA well.

This brings us to the discussion of the final component of uncertainty, which is the structural uncertainty of a model like FRISIA. With FRISIAv1.0, we have made a range of simplifications, the most important one being the high level of aggregation. Other major simplifications that impact the projected damages are the assumption of an evenly distributed population and assets in coastal segments, meaning that heterogeneity within a segment is completely neglected, or the use of just one possible type of flood protection, seawalls, where in reality the most suitable form of protection is highly dependent on local morphology. Furthermore, many of the cost formulations are highly simplified and uniform among all coastal zones, although we have approached this by assigning large uncertainties to parameters within these formulations

Apart from processes that are represented in a simplified way, there are also those processes that are missed entirely by FRISIAv1.0. One shortcoming is that FRISIAv1.0 does not account for local wave heights, which can have a major impact on realised damages, for example in a surge event. However, this is partly balanced by the high level aggregation, so that mean wave heights over the aggregated coastal zones are likely to be similar to some degree. Furthermore, our representation of damages accounts only for the direct damages to assets located in the coastal zone, but indirect damages to industries that rely on the coast, for example fisheries or tourism, are missed. Similarly, there is no information about the effect of compound events on damages, like extreme precipitation occurring together with a storm surge. It should be noted, however, that these processes are neglected by most models, especially when it comes to SLR impact models inside of IAMs.

## 7.2 Model application and improvements

As with all models, FRISIA is necessarily a simplification of reality. The most important limitation for FRISIAv1.0 is the high level of aggregation, which means that information about local morphology is only included implicitly in the susceptibility and annual exposure changes with SLR, but not in how morphology impacts local SLR damages or the feasibility of adaptation strategies. This precludes the use of FRISIA for investigating SLR impacts and damages in specific locations or supporting local decision making. Instead, FRISIA is a tool for simulating global or regional SLR impacts, and for understanding the large-scale socio-economic dynamics of SLR impacts in the coastal zone. In the previous section, we have laid out in detail the shortcomings of FRISIAv1.0 and the uncertainties that come with its results. Now we discuss the appropriateness of our model for the use in IAMs, which has been the main target application of FRISIA.

The main motivation for the development of FRISIA was to provide a computationally efficient model that can provide a diverse set of impact streams at a high level of aggregation. Running FRISIAv1.0 with a global aggregation until the year 2150 without making use of any parallelisation takes about 0.04 seconds on a standard laptop. This means that FRISIA can easily be coupled to any fast IAM without a considerable effect on computational efficiency, and large ensembles with a million members or more are feasible using parallel infrastructure. Apart from that, we have shown in this work that FRISIAv1.0 produces SLR impacts that are comparable to more complex models like CIAM, and we have incorporated new feedbacks to capture economic processes, like the diversion of investments away from exposed areas, that most other SLR impact models do not include. Lastly, we have demonstrated in Sect. 6 that coupling FRISIA to an IAM can lead to a broad range of interesting impacts of SLR, beyond a simple reduction in GDP. For these reasons, we conclude that FRISIAv1.0 will be a valuable tool for the IAM community, especially for dynamic IAMs that put focus on computational efficiency and global feedbacks.

When it comes to its use as a standalone model, the current version of FRISIA can only give global or regional insights about the large scale impacts of SLR. It can also be helpful for looking at more qualitative differences of how SLR impacts play out in coastal zones that are differentiated by other things than region, like we have separated out well protected from less protected regions in the bipolar setup applied in this study. Nevertheless, there are a range of model updates that should follow on, in order to improve the accuracy of estimated damages, foster FRISIA's usefulness as a standalone model of SLR impacts, and allow for more disaggregated setups in the future.

A major update to FRISIA will come from updating the underlying coastal database from DIVA (Vafeidis et al., 2008; Hinkel and Klein, 2009) to DSCIM-Coastal (Depsky et al., 2023). We did not base FRISIAv1.0 on DSCIM-Coastal mostly because our implementation of impacts is based on the CIAM model (Diaz, 2016; Wong et al., 2022), which uses DIVA as well. DSCIM-Coastal is a very recent framework and, even though Depsky et al. (2023) note that DSCIM-Coastal provides costs of SLR that are similar to those of CIAM, updating FRISIA with DSCIM-Coastal would improve the accuracy of the results, especially when going to more disaggregated setups. Major improvements that would come from this are the inclusion of more recent elevation data, heterogenous population and asset distributions within each segment, and a better resolution of coastal elevation.

There are also a range of other possibilities to improve FRISIA further, such as including a framework of wave height impacts on damages, updating local SLR weights, or better accounting for the effect of dike failures besides overtopping. A potential future model update that fits well with the focus on feedbacks in FRISIA is the inclusion of direct SLR impacts to coastal economies like fisheries and tourism, which could feed back to the evolution of coastal GDP. Other improvements, like including more options for coastal protection than just sea walls or creating a FRISIA version that operates on the coastal segment level, are conceivable, but more demanding. In the end, all further developments depend on the setting in which the model will be used, and the version of FRISIA presented here includes sufficient detail and processes to make it an informative modelling tool that is suitable for use in global or regional IAMs.

## 8   Conclusions

In this work, we presented FRISIA version 1.0, which is a "Feedback-based knowledge Repository for Integrated assessments of Sea level rise Impacts and Adaptation". FRISIA's module to calculate GMSL rise and its sub-components is largely based on the GMSL rise components of MAGICC (Nauels et al., 2017) and BRICK (Wong et al., 2017). The impacts and adaptation module of FRISIA is partly based on CIAM (Diaz, 2016; Wong et al., 2022), but is specifically designed for use in globally or regionally aggregated integrated assessment models that follow a non-equilibrium, dynamic modelling approach. In contrast to many other models for calculating SLR impacts, FRISIA integrates forward in time with a time step of one year, instead of following a cost-optimisation approach, therefore allowing for an easy implementation of coastal feedbacks and dynamics that have previously been neglected in coastal impact modelling. Nevertheless, FRISIA can in principle also be coupled to other types of IAMs, such as CGE models.

The presented no feedback version of FRISIA produces costs of SLR that are similar to those in the more comprehensive CIAM model (Wong et al., 2022). Incorporating additional coastal feedbacks - reduced growth in insufficiently protected coastal zones, and long-term negative effects of SLR-driven storm damages - shows that future SLR-driven storm damages might peak in the early 22$^{nd}$ century, despite continually increasing sea levels, simply because there will be less assets located in the susceptible coastal zone. It is worth pointing out that this does not happen because of planned retreat as a prescribed adaptation strategy, but rather in the case of no adaptation, as a consequence of potential economic feedback that are not included in other SLR impact components of IAMs. A similar effect has only been shown by a more complex, spatially disaggregated economic model before (Desmet et al., 2021). Furthermore, we have added a limit on the amount of money available for raising flood protection, which is typically not included in other SLR impact models. Adding such a limit means that some coastal regions will fail to fully protect or even build new flood protection at all in scenarios of extreme SLR in the future. The potential failure to further invest into flood protection is an important feedback that should be considered in adaptation decisions.

When coupling FRISIA to the new IAM FRIDA, which is developed within the same project as FRISIA and follows the same modelling strategy, a reduction of global GDP by $1.5 - 6.2$ % is observed for the mean SSP5-8.5 GMSL rise from the IPCC's AR6 report (0.77 m in 2100) and under a strategy with no adaptation, which is in the range of previous modelling

estimates (Hinkel et al., 2014; Brown et al., 2021; Cortés Arbués et al., 2024). Our simulations highlight the advantages of integrating a dynamic model like FRISIA into an IAM. This approach provides a diverse set of SLR impacts and adaptation costs that lead to more consequences than just a reduction in GDP, as, for example, the inflationary effects of no adaptation and protection strategies, respectively.

In summary, the here presented FRISIA model in its first published version 1.0 is a newly developed tool to study the socio-economic impacts of sea level rise in integrated assessment models. Due to the straightforwardness of the system dynamics modelling approach, new features can easily be added to the model in the future to explore more specific research questions. While, in terms of process-detail, FRISIA cannot compete with the implementation of SLR impacts in some other models (Stehfest et al., 2014; Rennert et al., 2022), it represents a major step forward for modelling the dynamic complexity of SLR and its impacts in IAMs. FRISIA's focus on feedbacks and time-dependent non-linear behaviour makes it ideal for coupling to dynamic IAMs that value dynamic complexity over process-detail. To our knowledge, FRISIAv1.0 represents the most comprehensive tool to study SLR impacts in this realm of models.

*Code and data availability.* The specific FRISIA and FRIDA model setups that were used in this manuscript, as well as all scripts and FRIDA output data needed to fully replicate the results are stored in a permanent repository on Zenodo (Ramme, 2025). FRISIA is maintained in a public GitHub repository at https://github.com/lnnrtrmm/FRISIA. The FRIDA model code is hosted on GitHub at https://github.com/metno/WorldTransFRIDA, whereas the full infrastructure to run ensembles with FRIDA is hosted at https://github.com/BenjaminBlanz/WorldTransFrida-Uncertainty.

*Author contributions.* Conceptualisation: LR, WAS, CS, CL. Funding acquisition: WAS, CS, CL. Investigation: LR, BB, CW. Methodology: LR, BB, CS, WAS, TEW. Software: LR, BB, WAS. Supervision: WAS, CS, CL. Visualisation: LR. Writing - original draft: LR. Writing - review and editing: BB, WAS, CW, TEW, CS, CL

*Competing interests.* The contact author has declared that none of the authors has any competing interests.

*Acknowledgements.* We thank Alexander Nauels for feedback on an early version of the model and Ben Callegari and Martin Grimeland for helpful discussions about the coupling of FRISIA output to the economy module of FRIDA. We further thank David Nielsen for internal review of the manuscript. This research was supported by the Horizon Europe research and innovation programs under grant agreement no. 101081661 (WorldTrans). This work used resources of the Deutsches Klimarechenzentrum (DKRZ) granted by its Scientific Steering Committee (WLA) under project IDs 33 and 1275.

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

  **Appendix A: SLR model description**

## A1  Thermal expansion

The ocean warms and expands under global warming, whereby it contributes the thermosteric component of GMSL rise ($\text{GMSL}_{\text{thermo}}$). The annual ocean heat uptake is an input parameter that can typically be provided within an IAM, and observational data of thermosteric GMSL rise show that it is strongly linked to estimated ocean heat content changes through the definition of a parameter called the expansion efficiency of heat (EEH, Marti et al., 2022). The EEH relates annual thermosteric GMSL rise to annual ocean energy uptake, which is the functional form we apply in the model:

$$\frac{d\text{GMSL}_{\text{thermo}}}{dt} = \text{EEH} \cdot \frac{d\text{OHC}}{dt}. \tag{A1}$$

There are several estimates for EEH in the literature, ranging from 0.12 m $\text{YJ}^{-1}$ (Levitus et al., 2012) to 0.145 and 0.15 m $\text{YJ}^{-1}$ (Marti et al., 2022; Church et al., 2011). Furthermore, comparing IPCC estimates of ocean energy input and thermosteric GMSL rise over the period 1971-2018 leads to an EEH estimate of 0.1177–0.1217 m $\text{YJ}^{-1}$ (Fox-Kemper et al., 2021, cross-chapter box 9.1), whereas we calculate a value of 0.0975 m $\text{YJ}^{-1}$ when fitting the observational data for $\text{GMSL}_{\text{thermo}}$ (Horwath et al., 2021) to estimates of ocean heat content change (Marti et al., 2022). However, choosing the most appropriate value of EEH in the SLR model also depends on how well the calculated ocean heat uptake fits with observational data. Hence, in our SLR model EEH is a calibration parameter, and we find that a range between 0.1 and 0.12 m $\text{YJ}^{-1}$ fits well with observational data and IPCC projections, when we force the SLR model with input from FaIR (see Fig. A1).

## A2  Land water storage

Changes in the Earth's stock of land water have had a significant contribution of 15 to 25 % to the observed total GMSL rise in the period 1993-2010, mainly driven by the depletion of groundwater and the loss of wetlands and forests (Reager et al., 2016). This is counteracted by the impoundment of water in dams for hydropower generation or irrigation purposes, which possibly resulted in a net fall of GMSL in the period 1950-1980, and which might have a significant impact in the coming decades as there is currently a major surge in dam construction in developing countries (Wada et al., 2017). Furthermore, climate variability dominates the decadal variability of land water storage (LWS) through processes like El Niño (Reager et al., 2016), but the long term effect of climate change is unclear (Jensen et al., 2019). Similarly, the future trends of groundwater depletion, forest and wetland loss or freshwater impoundment by dams are highly uncertain (Wada et al., 2017).

Corresponding to the unknown future developments, the BRICKv0.2 and MAGICC6 models simply assume a more or less constant increase of the GMSL rise contribution of land water storage changes ($\text{GMSL}_{\text{LWS}}$, Wong et al., 2017; Nauels et al., 2017). Alternatively, $\text{GMSL}_{\text{LWS}}$ can also be approximated by a semi-empirical approach relating groundwater extraction and hydropower dam building to global population (Rahmstorf et al., 2012; Kopp et al., 2014). As we design our SLR model for the use in IAMs, which in most cases can provide population data endogenously, we follow Rahmstorf et al. (2012) and relate

the rate of $\text{GMSL}_{\text{LWS}}$ change linearly to global population:

$$\frac{\text{dGMSL}_{\text{LWS}}(t)}{\text{d}t} = c \cdot \text{Pop}(t). \tag{A2}$$

Varying the population sensitivity $c$ within $1 \cdot 10^{-8}$ to $6 \cdot 10^{-8}$ m Mp$^{-1}$ yr$^{-1}$ approximately fits with observational data and gives $\text{GMSL}_{\text{LWS}}$ projections that are in line with IPCC estimates (see Fig. A2). In the case that no population data is given, the model assumes a constant rate between 0.0002 and 0.0004 m yr$^{-1}$, which gives a comparable fit to observational data and IPCC estimates. It should be noted that some of the more process-based IAMs directly simulate the land processes that cause changes in GMSL, such as groundwater withdrawal or the construction of hydropower dams. In this case, $\text{GMSL}_{\text{LWS}}$ can be calculated directly, which is the approach followed in the FRISIA version implemented in the FRIDA IAM (Schoenberg et al., 2025).

## A3 Mountain glaciers

For the contribution of the melting of mountain glaciers to GMSL rise ($\text{GMSL}_{\text{MG}}$), we adopt the parameterization from Wigley and Raper (2005) that is used in a range of other models as well (Perrette et al., 2013; Wong et al., 2017; Li et al., 2020). The formulation relates the current rate of $\text{GMSL}_{\text{MG}}$ to the GSAT anomaly ($T$) and the remaining fraction of the GMSL rise potential of mountain glaciers ($V_0 = 0.41$ m):

$$\frac{d\text{GMSL}_{\text{MG}}}{d\text{t}} = \beta_0 \cdot T^p \cdot (1.0 - \frac{\text{GMSL}_{\text{MG}}}{V_0})^n. \tag{A3}$$

The exponent $n = 1.646$ is the scaling coefficient between global glacier area and volume (Wigley and Raper, 2005; Perrette et al., 2013), and we introduced the nonlinearity exponent $p = 1.5$ to the GSAT anomaly to improve the fit with IPCC AR6 estimates. The sensitivity parameter $\beta_0$ is varied between $0.4 \cdot 10^{-3}$ and $1.0 \cdot 10^{-3}$ m $°C^{-1.5}$yr$^{-1}$ to match the estimated uncertainty of projections in AR6 (see Fig. A3).

## A4 Greenland ice sheet

The Greenland ice sheet contribution to GMSL rise ($\text{GMSL}_{\text{GrIS}}$) arises from two processes (Goelzer et al., 2020). The first of these is the change in the surface mass balance (SMB) under global warming, which effectively means increased surface melting and runoff in summer, counteracted by increased snowfall in winter. In the second process, $\text{GMSL}_{\text{GrIS}}$ is driven by changes in the solid ice discharge (SID) to the ocean, which increases as the surface ocean warms. The SMB component is estimated to be dominating $\text{GMSL}_{\text{GrIS}}$, but uncertain high impact processes could add significantly to this or the SID component (Bassis et al., 2021; Fox-Kemper et al., 2021). Here, we choose to adopt the formulation that is used in MAGICC6 (Nauels et al., 2017), because it is only driven by the GSAT anomaly and explicitly tracks changes in both components with

$$\text{GMSL}_{\text{GrIS}} = \text{GMSL}_{\text{GrIS}}^{\text{SMB}} + \text{GMSL}_{\text{GrIS}}^{\text{SID}} \tag{A4}$$

and

$$\frac{d\text{GMSL}_{\text{GrIS}}^{\text{SMB}}}{d\text{t}} = \nu T^{\varphi} \cdot \left(1 - \frac{\text{GMSL}_{\text{GrIS}}^{\text{SMB}}}{\text{GMSL}_{\text{GrIS}}^{\text{SMB,max}}}\right)^{0.5}, \tag{A5}$$

where $\text{GMSL}_{\text{GrIS}}^{\text{SMB,max}} = 7.36$ m is the maximum contribution of the SMB component to GMSL rise (Bamber et al., 2013) and $\varphi = 2.0$ is the non-linearity of the temperature relationship. These parameter values and the fact that there is only a non-linear temperature term are in line with the default settings in MAGICC6 (Nauels et al., 2017). We use the temperature sensitivity $\nu$ to add uncertainty to the SMB component by varying it between $0.5 \cdot 10^{-4}$ and $2.0 \cdot 10^{-4}$ m $°\text{C}^{-2}\text{yr}^{-1}$.

We modify the MAGICC6 formulation of the SID component by adding a switch to include a high impact behaviour, where processes with a potential large contribution to GMSL rise add to this component above a certain temperature threshold. This is implemented as a constant fraction $R_{\text{GrIS}}$ of the remaining part of the maximum GMSL rise contribution of the SID component, $\text{GMSL}_{\text{GrIS}}^{\text{SID,max}}$, and this fraction is added to the general SID term, if a critical value of the GSAT anomaly is crossed:

$$\frac{d\text{GMSL}_{\text{GrIS}}^{\text{SID}}}{dt} = \max\left(0, \left(\text{GMSL}_{\text{GrIS}}^{\text{SID,max}} - \text{GMSL}_{\text{GrIS}}^{\text{SID}}\right)\left(\varrho e^{\epsilon T} + R\right)\right) \tag{A6}$$

$$\text{with} \quad R = \begin{cases} R_{\text{GrIS}}, & \text{if } T > T_{\text{crit}} \\ 0, & \text{else} \end{cases}. \tag{A7}$$

The original formulation in MAGICC6 is based on previous work that projected the outflow of the four main outlet glaciers, which contribute approximately 22% to the overall solid ice discharge of Greenland (Nick et al., 2013), hence the calculated $\text{GMSL}_{\text{GrIS}}^{\text{SID}}$ was multiplied with a factor of five in the end (Nauels et al., 2017). Here, we simplify this equation, without changing it qualitatively, by incorporating the factor five already into the the maximum GMSL rise contribution of the SID component $\text{GMSL}_{\text{GrIS}}^{\text{SID,max}}$, for which we use the maximum value of 0.42 m suggested by Winkelmann and Levermann (2013), noting that the total amount might be significantly smaller. In order to account for that, our choice of value for the temperature sensitivity $\epsilon = 0.39$ in this component is that of the minimum case calibrated in Nauels et al. (2017), so that the contribution of $\text{GMSL}_{\text{GrIS}}^{\text{SID}}$ is only substantial during the 21$^{\text{st}}$ and 22$^{\text{nd}}$ century, if the high impact switch is turned on. We account for uncertainty in the SID component via several parameters. The sensitivity of the temperature dependent discharge fraction $\varrho$ is varied between $10^{-4}$ and $5 \cdot 10^{-4}$ yr$^{-1}$. At activated high impact behaviour, the constant discharge fraction above the temperature threshold $R_{\text{GrIS}}$ is varied between $10^{-3}$ and $10^{-2}$ yr$^{-1}$, signifying that between $0.1 - 1\%$ of the remaining GMSL rise potential of the SID component is removed annually through uncertain high impact processes, if this feature is activated. Additionally, we vary the critical threshold value of the GSAT anomaly above which the high impact behaviour is triggered between 2.5 and 4 $°\text{C}$. Altogether, this formulation for $\text{GMSL}_{\text{GrIS}}$ combined with the given parameter ranges matches well with the projections from AR6 and the likely ranges given therein (see Fig. A4). Only for the high-end emissions scenario SSP5-8.5 this formulation leads to projections that reach below the likely range in AR6 and do not cover the highest estimates in the "Low Confidence" scenario, even with activated high impact behaviour.

## A5  Antarctic ice sheet

The Antarctic ice sheet contribution to global sea level rise ($\text{GMSL}_{\text{AntIS}}$) was minor during the era of satellite observations, potentially exhibiting even an ice sheet growth in East Antarctica (Bell and Seroussi, 2020; Stokes et al., 2022). However, recently there have been increasing observations of ice sheet decline, hinting to a potentially substantial acceleration of $\text{GMSL}_{\text{AntIS}}$ un-

der global warming (Pattyn and Morlighem, 2020). Additionally, uncertain processes of rapid ice sheet disintegration could lead to even higher rates of mass loss, but too little is known about their true potential (Noble et al., 2020). Generally, the potential of the Antarctic ice sheet is by far the largest of all contributions to GMSL rise (Morlighem et al., 2020), and just like for the Greenland ice sheet, $GMSL_{AntIS}$ is also driven by changes in the surface mass balance and solid ice discharge under climate change.

Some existing models for the estimation of $GMSL_{AntIS}$ have assumed simple linear relationships to the GSAT anomaly (Wigley, 2018; Li et al., 2020), while more detailed approaches for example model the SMB component and the SID of the different sectors of the Antarctic ice sheet separately (Nauels et al., 2017). Here, we choose to adopt the strategy of the BRICKv0.2 model (Wong et al., 2017), which applies the Danish Center for Earth System Science Antarctic Ice Sheet (DAIS) model (Shaffer, 2014). The DAIS model simulates the Antarctic ice sheet as an axisymmetric structure and tracks its volume and radius explicitly. It can be easily calibrated to produce a wide range of $GMSL_{AntIS}$ future projections. The model formulation includes a large set of parameters and equations, which is why we do not show them here, but refer the reader to the original publication (Shaffer, 2014). Vega-Westhoff et al. (2019) present a comprehensive calibration of the BRICK model parameters, including those from the DAIS model, and show the bounds and uncertainty ranges of those parameters. Here, we mostly adopt the parameter values of the original parameterization in Shaffer (2014), and adjusted a few parameters to fit the model projections to historical observations and IPCC AR6 estimates (see Fig. A5). We further follow Wong et al. (2017) and calculate the Antarctic surface air and Antarctic ocean temperatures, which are needed in DAIS, through simple relationships with the GSAT anomaly. Lastly, we implement a switch for a high impact behaviour parameterization that is also applied in a newer version of BRICK (Wong et al., 2022), which adds a constant rate of $4 \pm 4$ mm yr$^{-1}$ to $GMSL_{AntIS}$ if the Antarctic air temperature crosses a critical threshold of $14.5 \pm 0.5°$C, which corresponds to a global mean temperature anomaly of $2.7 - 3.5°$C . Just like for the other components, ranges are provided to two parameters of the DAIS model, as well as the constant disintegration rate above the high impact behaviour threshold and the critical Antarctic air temperature, to let the uncertainty range of the model projection match the likely ranges presented in AR6 (see Fig. A5).

**Appendix B:  Additional material**

**Table A1.** Overview of the uncertainty parameters in the SLR module of FRISIAv1.0. See the corresponding appendix sections for more explanation of the parameters and Sect. 3.1 for the discussion of how these parameters are varied.

| component | parameter | lower | upper | unit | equation | comment |
|---|---|---|---|---|---|---|
| $\text{GMSL}_{\text{thermo}}$ | EEH | 0.1 | 0.12 | m YJ$^{-1}$ | Eq. A1 | |
| $\text{GMSL}_{\text{LWS}}$ | $c$ | $10^{-8}$ | $6 \cdot 10^{-8}$ | m MP$^{-1}$ yr$^{-1}$ | Eq. A2 | if population data is available |
| $\text{GMSL}_{\text{LWS}}$ | $r_{\text{LWS}}$ | $2 \cdot 10^{-4}$ | $2 \cdot 10^{-4}$ | m yr$^{-1}$ | - | if population data is not available |
| $\text{GMSL}_{\text{MG}}$ | $\beta_0$ | $4 \cdot 10^{-4}$ | $10^{-3}$ | m $°\text{C}^{-1.5}$ yr$^{-1}$ | Eq. A3 | |
| $\text{GMSL}_{\text{GrIS}}$ | $\nu$ | $0.5 \cdot 10^{-4}$ | $2.0 \cdot 10^{-4}$ | m $°\text{C}^{-2}$ yr$^{-1}$ | Eq. A5 | |
| $\text{GMSL}_{\text{GrIS}}$ | $\rho$ | $10^{-4}$ | $5 \cdot 10^{-4}$ | yr$^{-1}$ | Eq. A6 | |
| $\text{GMSL}_{\text{GrIS}}$ | $T_{crit}$ | 4 | 2.5 | $°\text{C}$ | Eq. A7 | higher value means crossing the threshold later |
| $\text{GMSL}_{\text{GrIS}}$ | $R_{GrIS}$ | $10^{-3}$ | $10^{-2}$ | yr$^{-1}$ | Eq. A7 | constant disintegration fraction after crossing |
| $\text{GMSL}_{\text{AntIS}}$ | $a_{anto}$ | 0.15 | 0.3 | $°\text{C} \, °\text{C}^{-1}$ | - | Wong et al. (2017, Eq. 6) |
| $\text{GMSL}_{\text{AntIS}}$ | $h_0$ | 1600 | 1900 | m | - | Shaffer (2014, Table 1) |
| $\text{GMSL}_{\text{AntIS}}$ | $T_{crit}^{Ant}$ | -14.0 | -15.0 | $°\text{C}$ | - | threshold for high impact behaviour |
| $\text{GMSL}_{\text{AntIS}}$ | $R_{AntIS}$ | 0 | 0.008 | m yr$^{-1}$ | - | constant disintegration rate after crossing |

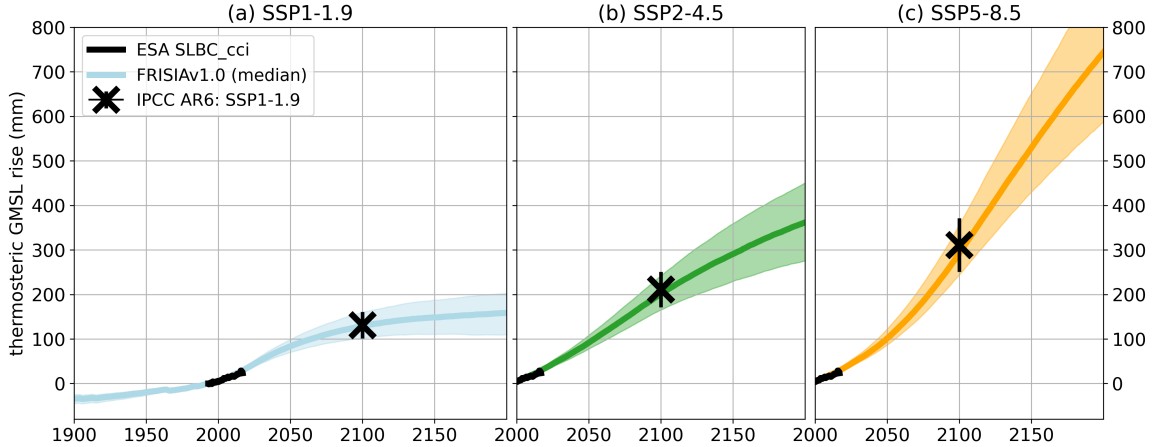

**Figure A1.** Contribution of the thermosteric component of GMSL rise compared to data of the European Space Agency (ESA) Sea Level Budget Closure - Climate Change Initiative (SLBC_CCI) project (Horwath et al., 2021, black curve). Given are also the median estimates for 2100 from the Intergovernmental Panel on Climate Change's (IPCC) Sixth Assessment Report (AR6) for SSP1-1.9 (a), SSP2-4.5 (b) and SSP5-8.5 (c). Given error bars are the "likely ranges" from Table 9.9 in Fox-Kemper et al. (2021). Correspondingly, we show also the median as the solid line and the $17^{\text{th}} - 83^{\text{rd}}$ percentile range as the shaded area for the data from FRISIA.

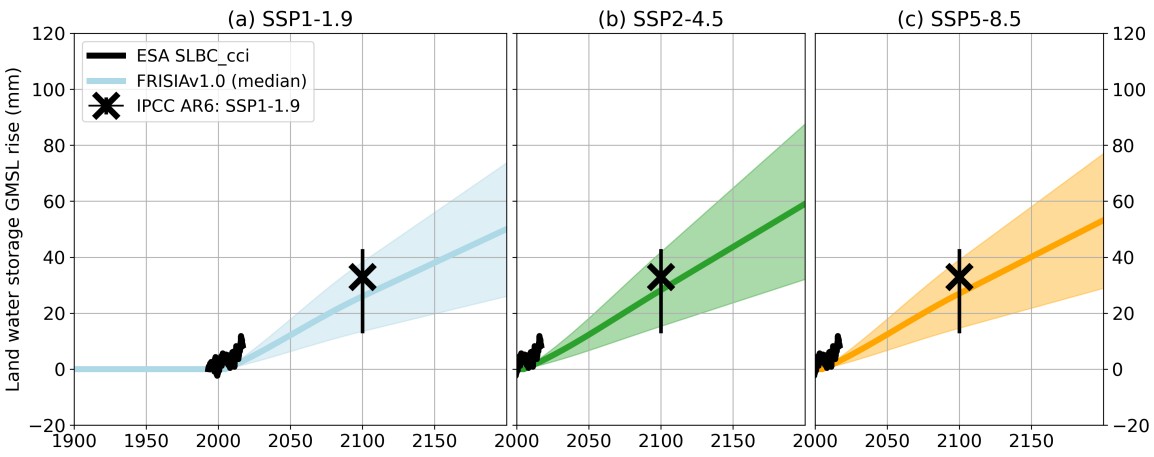

**Figure A2.** Same as Fig. A1, but for the GMSL contribution of land water storage changes.

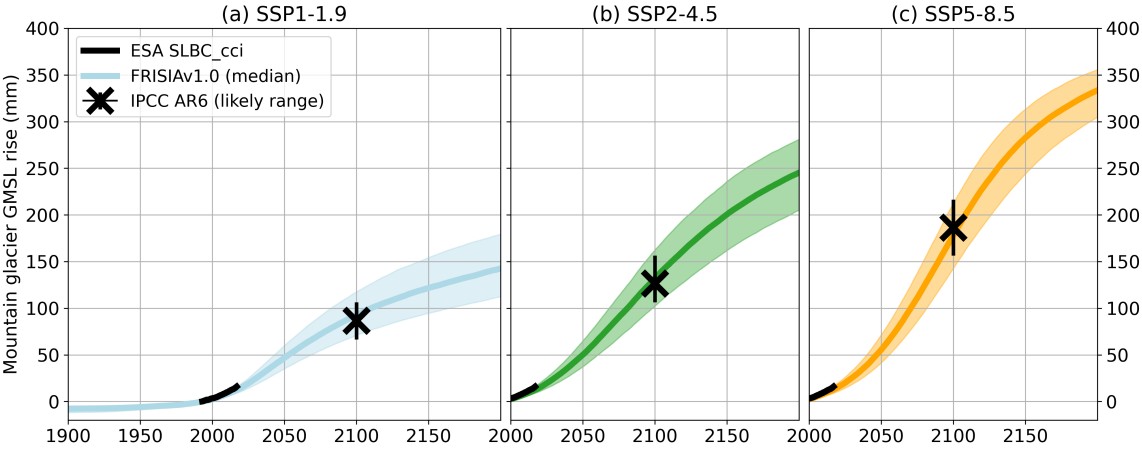

**Figure A3.** Same as Fig. A1, but for the GMSL contribution of mountain glaciers.

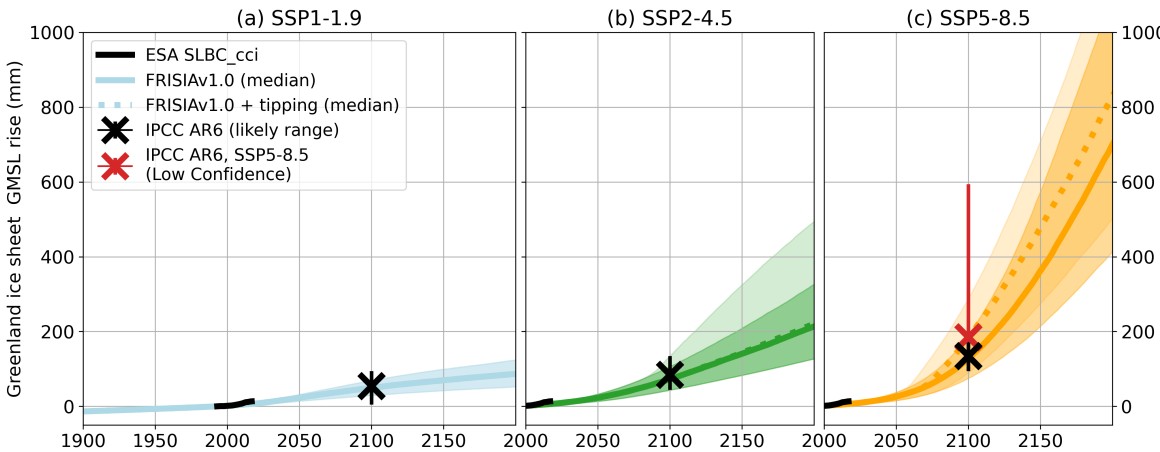

**Figure A4.** Same as Fig. A1, but for the GMSL contribution of the Greenland ice sheet. Here, we present also the "Low Confidence" scenario as the red error bar for SSP5-8.5, which includes processes with a potentially high impact in which there is low confidence (Fox-Kemper et al., 2021). Therefore, we also show two curves and the corresponding uncertainty ranges of FRISIA, whereby the solid and dashed curves represent cases with deactivated and activated high impact behaviour parameterization in FRISIA, respectively.

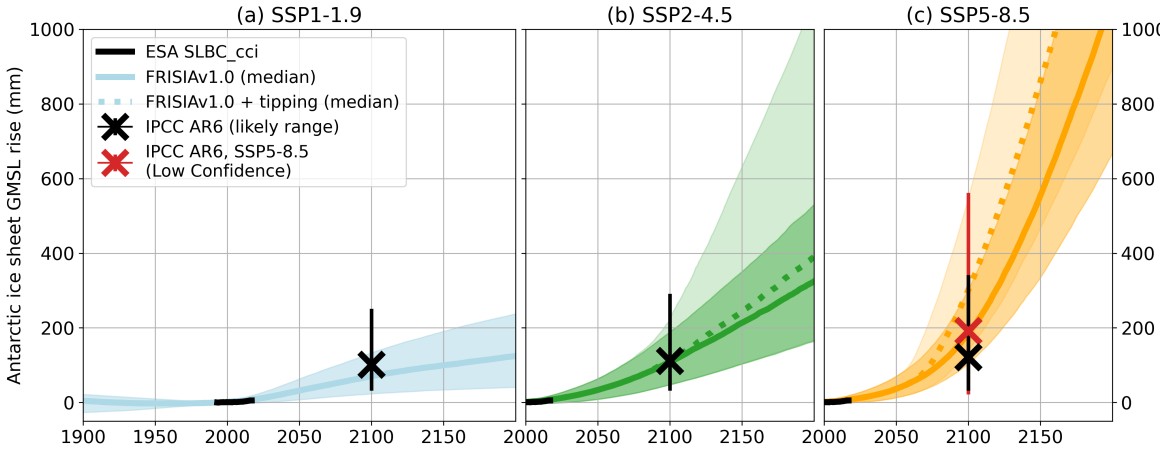

**Figure A5.** Same as Fig. A1, but for the GMSL contribution of the Antarctic ice sheet, including the possible contribution of high impact behaviour as in Fig. A4.

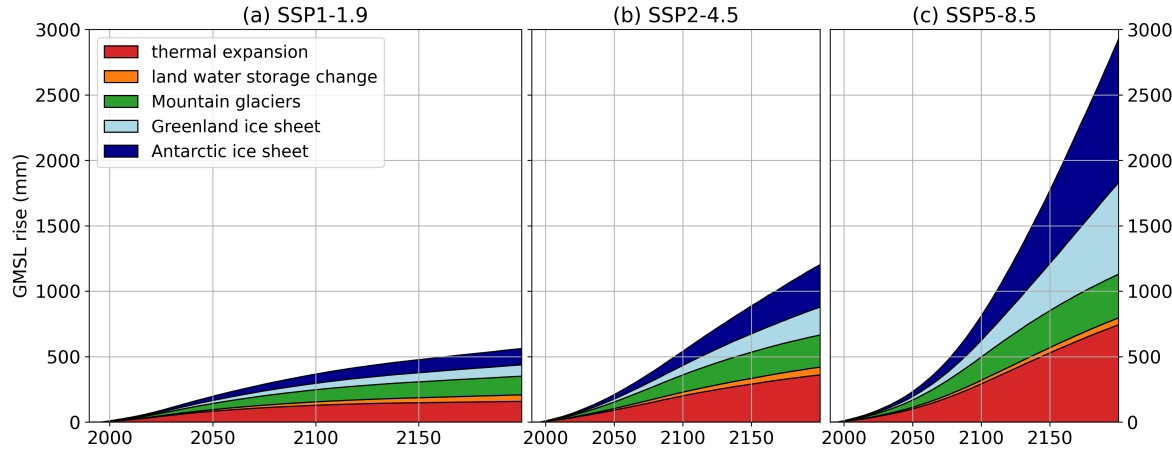

**Figure A6.** Same as previous figures, now showing the median estimate (without the high impact behaviour contribution) of all components at once.

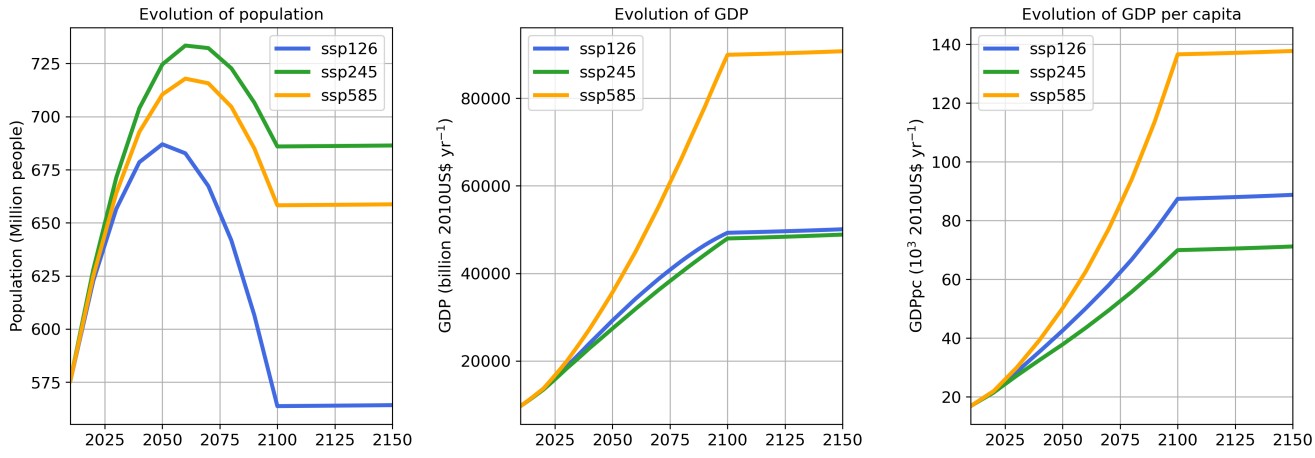

**Figure A7.** The evolution of global coastal population, GDP and GDP per capita, as used for the "global" aggregation of the FRISIA model. See Sect. 3.2.1 on how the data is constructed.

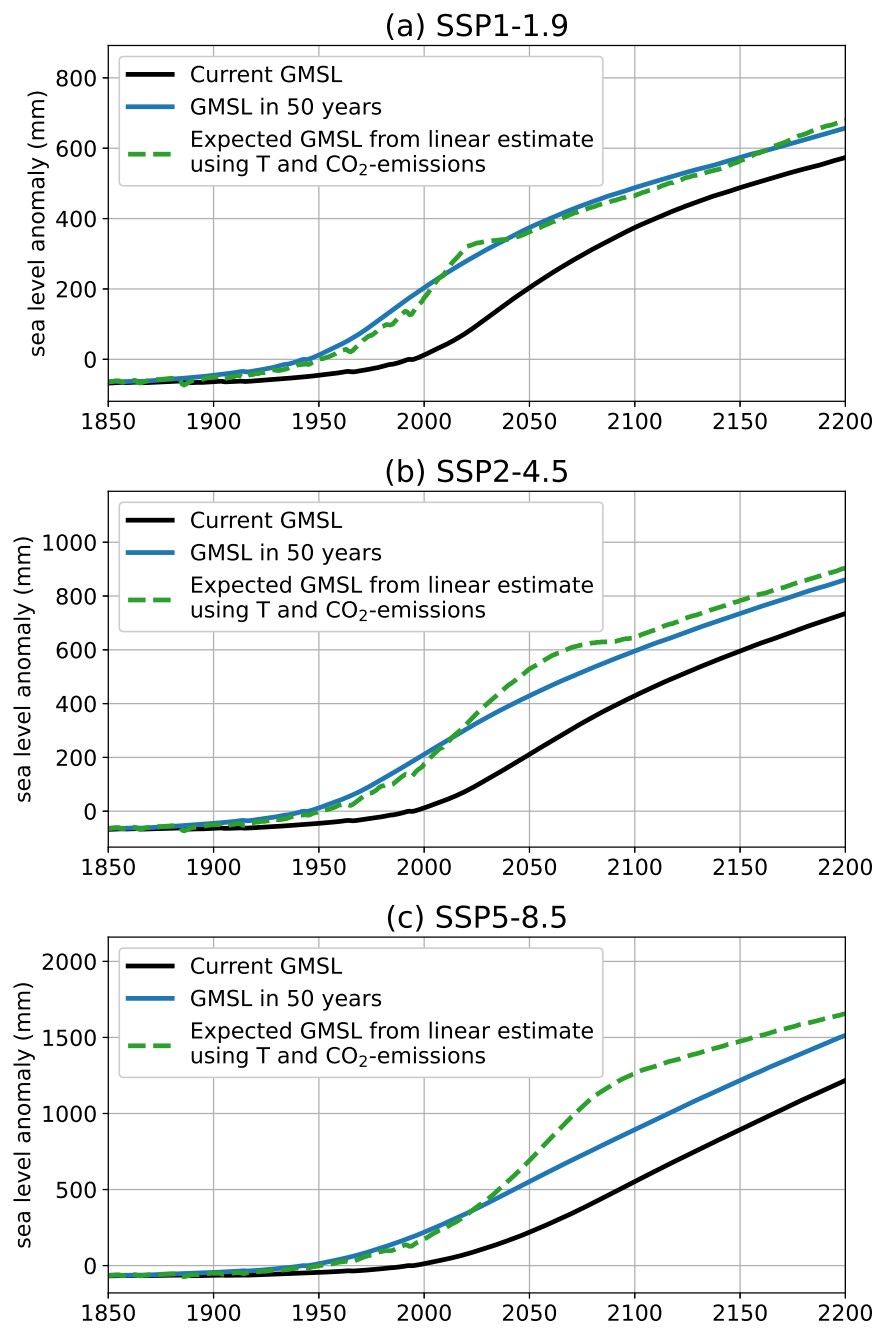

**Figure A8.** The expected GMSL rise in 50 years is fitted using the current GSAT anomaly and $CO_2$ emissions as predictors in $GMSL_{50} = GMSL_{today} + a \cdot T_{today} + b \cdot emisCO_2$, with a=0.091 m $K^{-1}$ and b=0.0131 m $(Gt\ C)^{-1}$ for SSP1-1.9 (a), SSP2-4.5 (b) and SSP5-8.5 (c).

**Table A2.** Information on coastal segments from the DIVA database aggregated in the three different ways discussed in this study: 1. Global aggregation, 2. Bipolar aggregation, separated by whether the flood protection height, $H$, in a segment is higher than the height of a storm surge with a return period of 1000 years, s1000. 3. Regional aggregation, separated by World Bank regions. Population numbers are taken from the DIVA database and values of GDP are calculated using the country-specific GDP per capita values for 2010 from IIASA data (Riahi et al., 2017; Cuaresma, 2017) The local SLR factor, $SLR_{fac}$, is calculated by fitting a linear relationship between the population weighted GMSL rise for every aggregation level and calculated GMSL rise. However, this is not used by FRISIA, which instead uses aggregated weights for the individual GMSL components, as this approach gives more reliable results when the relative contribution of the GMSL components changes compared to the data used for the fitting. Nevertheless, $SLR_{fac}$ gives a good indication of the relative local SLR for specific regions or aggregation levels, for example that relative local SLR will be significantly larger in the East Asia and Pacific region than in the Europe and Central Asia region.

| Aggregation level | segments # | coastline length $10^3$ km (%) | population Mp (%) | GDP $10^9$\$ yr$^{-1}$ (%) | GDP per capita $10^3$\$ yr$^{-1}$ p$^{-1}$ | $\overline{H}$ m | $SLR_{fac}$ |
|---|---|---|---|---|---|---|---|
| Global | 12148 | 1038.2 (100) | 575.6 (100) | 9693 (100) | 16.8 | 1.94 | 0.984 |
| H < s1000 | 8459 | 662.9 (63.8) | 342.6 (59.5) | 2371 (24.5) | 6.9 | 1.46 | 1.038 |
| H > s1000 | 3689 | 375.3 (36.2) | 233.0 (40.5) | 7323 (75.5) | 31.4 | 2.80 | 0.930 |
| North America | 1772 | 304.0 (29.3) | 54.5 (9.5) | 2505 (25.8) | 46.0 | 3.59 | 0.930 |
| Latin America and Caribbean | 1258 | 100.1 (9.6) | 59.5 (10.3) | 708 (7.3) | 11.9 | 1.93 | 1.039 |
| Europe and Central Asia | 3286 | 388.9 (37.5) | 89.7 (15.6) | 2765 (28.5) | 30.8 | 2.48 | 0.829 |
| Middle East and North Africa | 383 | 21.3 (2.1) | 26.1 (4.5) | 259 (2.7) | 9.9 | 1.62 | 0.956 |
| Sub-Saharan Africa | 590 | 34.5 (3.3) | 40.2 (7.0) | 92 (1.0) | 2.3 | 0.95 | 1.036 |
| South Asia | 750 | 15.2 (1.5) | 70.1 (12.2) | 234 (2.4) | 3.3 | 1.7 | 1.037 |
| East Asia and Pacific | 4109 | 174.0 (16.8) | 235.6 (40.9) | 3131 (32.3) | 13.3 | 2.05 | 1.075 |

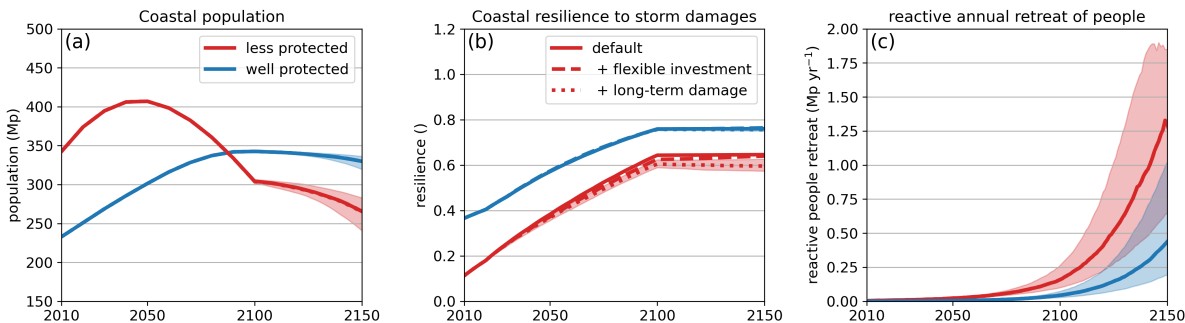

**Figure A9.** Additional variables for the experiments from Sect. 5.1 and Fig. 6. Here we show how coastal population (a), storm damage resilience (b) and the reactive retreat of people in response to inundation (c) evolve with and without activated economic feedbacks. The evolution of coastal GDP is directly proportional to that of coastal assets in Fig. 6 (a), and reactive assets retreat follows a similar trajectory to that of people in panel (c) of this figure.

**Table A3.** Overview of the uncertainty parameters in the impacts and adaptation module of FRISIAv1.0.

| parameter | parameter name in source code | unit | default | min | max | reference/comment |
|---|---|---|---|---|---|---|
| $C_{\text{construct,ref}}$ | fp_construction_cost_reference | b\$/m$^2$/km | 0.00602 | 0.005 | 0.007 | CIAM (Wong et al., 2022) |
| $f_{\text{maintain}}$ | maintenance_cost_fraction | year$^{-1}$ | 0.02 | 0.015 | 0.03 | CIAM (Wong et al., 2022) |
| $f_{\text{land}}$ | land_opportunity_cost_rate | year$^{-1}$ | 0.04 | 0.03 | 0.05 | CIAM (Wong et al., 2022) |
| $\tau_{\text{protect}}$ | fp_construction_duration | year | 10 | 5 | 25 | |
| $f_{\text{invest}}$ | maximum_gdp_fraction_for_fp_investment | 1 | 0.03 | 0.01 | 0.05 | |
| $\rho_{\text{threshold}}$ | safe_coastal_zone_likelihood_threshold | 1 | 0.95 | 0.9 | 1.0 | |
| $S_{50,\text{halfpoint}}$ | effective_flood_height_at_which_investment_is_halved | m | 1.0 | 0.5 | 3.0 | |
| $f_{\text{maxDamage}}$ | flood_event_damage_fraction | 1 | 0.3 | 0.2 | 0.4 | |
| $f_{\text{repair}}$ | fraction_of_storm_damages_that_is_repaired | 1 | 0.9 | 0.75 | 1.0 | |
| $\tau_{\text{retreat}}$ | proactive_retreat_time_scale | year | 10 | 5 | 25 | |
| $f_{\text{fatality}}$ | flood_event_fatality_rate | 1 | 0.01 | 0.005 | 0.02 | CIAM (Wong et al., 2022) |
| $lv(t=0)$ | coastal_land_value_init | b\$ km$^{-2}$ | 0.005376 | 0.005 | 0.006 | CIAM (Wong et al., 2022) |
| $\gamma$ | increase_factor_for_costs_of_reactive_retreat | 1 | 4 | 3 | 5 | CIAM (Wong et al., 2022) |
| $c_{\text{demolition}}$ | asset_demolition_cost_factor | 1 | 0.05 | 0.025 | 0.075 | CIAM (Wong et al., 2022) |
| $f_{\text{mobile}}$ | mobile_asset_fraction | 1 | 0.25 | 0.2 | 0.3 | CIAM (Wong et al., 2022) |
| $c_{\text{relocate,A}}$ | asset_relocation_cost_factor | 1 | 0.1 | 0.05 | 0.15 | CIAM (Wong et al., 2022) |
| $f_{\text{remaining}}$ | not_depreciated_fraction_of_assets_at_time_of_retreat | 1 | 0.1 | 0 | 0.2 | |

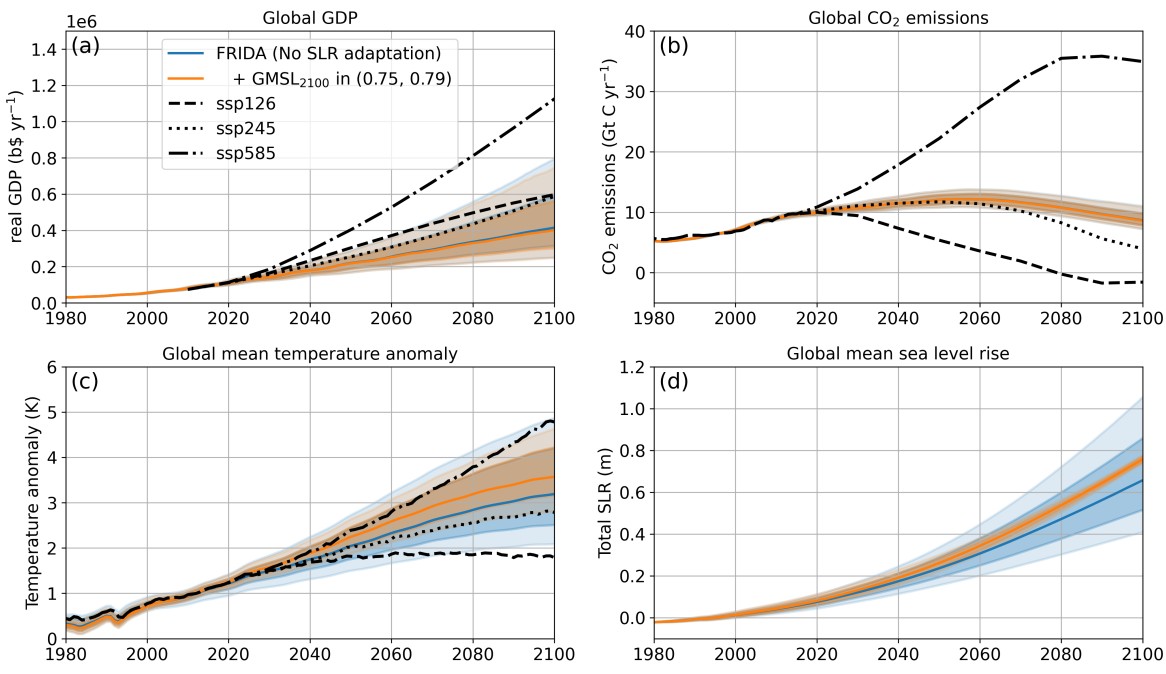

**Figure A10.** Output of the model uncertainty analysis that ran 20,000 simulations of FRIDA v2.1, varying the uncertainty parameters of the model using the Sobol sequence approach (Sobol, 1967). The blue line and shaded area depict the median and the $17^{th} - 83^{rd}$ percentile range of the full uncertainty ensemble. The orange line and shaded area depict the median and $17^{th} - 83^{rd}$ percentile range of those simulations of the full ensemble that lead to a GMSL rise of 0.75-0.79 m by 2100 (the median response in the SSP5-8.5 scenario). The black lines represent the temperature and GDP data that were used to force the offline version of FRISIA, as described in the main text. It can be seen that the FRIDA model is approximately comparable to the SSP2-4.5 scenario in global mean temperature and $CO_2$ emissions.