# Peer review of "Feedback-based sea level rise impact modelling for integrated assessment models with FRISIAv1.0"

_EGUsphere, 2025_

## Author Comment (AC1)

We thank both reviewers for their constructive feedback. We have made revisions according to each comment and believe that this has improved the manuscript substantially. Please see below for the specific responses to each comment.

Reviewer's comment
Author's response
- *Change in manuscript (given line numbers refer to the **manuscript with tracked changes**)*

**Anonymous Reviewer #1**

In this paper, the authors present FRISIA 1.0, a spatially highly aggregated model of sea-level rise impacts on coastal assets, population, and adaptation costs. The paper builds upon the prior CIAM model (Diaz, 2016; Wong et al., 2022) by incorporating impacts on asset prices as well as on population and adaptation, thus providing a feedback mechanism by which the coastal impacts of sea-level rise can affect coastal assets beyond the effects of relocation costs. The model is calibrated against CIAM, but whereas CIAM considers benefits and costs along 12,000+ coastal segments, this model – designed for use in highly aggregated integrated assessment models – operates with 1-7 aggregate regions.

We thank the reviewer for the overall very constructive feedback.

**Major comments**

***Deeper comparison to the literature:*** Overall, I like this paper, but find it could benefit from a deeper comparison to the literature. Most especially, the effects on asset prices seem ex nihilo; there are no references to economic theory papers justifying the formalism, nor any comparison to the growing econometric literature (e.g., see review by Contat et al., 2024 on the empirical relationship between climate risk and real estate prices).

We thank the reviewer for the helpful reference to the review of Contat et al. (2024). Following the assumption that with "effects on assets" the reviewer is referring to the new feedback on assets that we added in FRISIA compared to CIAM, that is, their reduced growth rate under expected flooding and the potential long-term reduction of asset values from increased storm surge damages, we have added more explanation and references to the corresponding parts in the model description section (now Sect. 3.2.4).

*- new text at lines 387-391, 397-399 and 430-434 in Sect. 3.2.4*

The authors would also benefit by a comparison to Depsky et al. (2023), who construct a coastal dataset similar in structure to DIVA but with updated inputs and then build a CIAM-like model in Python with improved representation of several processes and a data-informed estimate of the relocation cost parameter. Similar to the use of CIAM in the GIVE model described in the manuscript, the "DSCIM-Coastal" model of Depsky et al. (2023) was used for the same recent Social Cost of

Greenhouse Gas (SC-GHG) estimation exercise as part of the Data-driven Spatial Climate Impact Model (DSCIM, Climate Impact Lab 2023). DSCIM-Coastal contains an updated and open-source alternative to the DIVA input data used by the authors, as well as an updated Python implementation of the model that the authors modify and reimplement in Python.

This is indeed a very relevant publication, and we thank the reviewer for bringing it to our attention. In fact, we believe that in the future we can base our model on this database rather than the older DIVA database. For the manuscript under review, however, this would mean to lose the basis for our comparison to the CIAM data, and require a substantial amount of work, so that for now we remain with using DIVA as a basis.

We now properly introduce the DSCIM-Coastal model in the background section 2 and elaborate on the potential use of it in a future version of FRISIA also in the new discussion section that we added in response to this and other comments.

*- New and updated text at lines 76-84 in Sect. 2*
*- Reference to discussion section at lines 185-187 in Sect. 3.2*
*- New paragraph at lines 922-928 in new Sect. 7*

The authors would also benefit by a comparison to the more theoretically grounded dynamic spatial integrated assessment model of Desmet et al. (2021), who do not explicitly account for protection but do model flows of capital and population in response to sea-level rise, and like this paper show losses in exposed areas peaking and declining over time.

Again, we thank the reviewer for this valuable reference. We now cite this paper along with other papers of economic damages from SLR in the background section, and return to it when discussing the peak shape of our modelled damages.

*- New text al lines 60-62 in Sect. 2*
*- New paragraph at lines 710-714 in Sect. 5.1*
*- New text at lines 953-954 in Sect. 8*

***Appropriateness of CIAM parameterization:*** FRISIA applies formalism similar to those of CIAM, but rather than operating at a highly resolved coastal segment scale, it applies them at a global or World Bank-region scale. The appropriateness of doing this could be better justified. (Maybe this justification is simply – we are modeling for insights rather than for numbers, and we are fine with our flooding costs being within a factor of X of CIAM results.) As an integration test, the comparison to CIAM in Figures 3-4 does not always give the greatest confidence that FRISIA is a good emulator of CIAM (e.g., compare relocation costs in Figure 3).

To the extent the goal is to emulate the results of CIAM (after adjustment for the difference in the treatment of initial flood protection height), some quantitative performance metrics might be useful.

It is true that the focus of a model like FRISIA is rather on general insights about the global economic effect of SLR and getting the global costs approximately right (or rather: appropriately covering the uncertainty spread), and the model cannot provide advice for specific coastal regions. In that regard it is different from CIAM, and we do not intend to simply emulate the results of CIAM. CIAM builds the

basis for our model, but through the incorporation of new feedbacks, which are not represented in CIAM, we specifically try to capture new dynamics that might change the projected numbers. This means that we build FRISIA around those parts of CIAM that are useful for our model, but incorporate new formulations, where we think there is room for improvement.

Despite the fact that differences to CIAM were already discussed in the previous version of the manuscript (e.g. lines 412-440 in the old version of the manuscript), we agree that this topic can be discussed in even more detail. As we added a new section on inputs to the impacts and adaptation module in response to other comments, we use that space now to also discuss the appropriateness of the use of CIAM parametrisations. We now also discuss the differences between CIAM and FRISIA in the new discussion section.

Lastly, regarding the specific difference in relocation costs in Fig. 4 (g-i), it is questionable that the CIAM results do actually provide a reasonable temporal evolution, because of the way that costs are calculated over time in the CIAM model. For example, global mean SLR has no big variation between the scenarios in the year 2050 yet. So, even when accounting for the fact that GDP is higher by about 20-30% in SSP5-8.5 in 2050, why should relocation costs be higher by a factor of 4-5 in that year already? Similarly, why should relocation costs almost triple between 2080 and 2090 in SSP2-4.5? In this regard, we believe that the FRISIA projections draw a more consistent picture in terms of the temporal evolution of costs. That said, it is true that a discussion of this difference is useful, and we added a paragraph to that section.

*- text at lines 228-232 in new model input Sect. 3.2.1*
*- note on the use of CIAM formulation for flood protection at lines 332-337 in Sect. 3.2.3*
*- new sentence in the CIAM comparison at lines 577-579 in Sect. 4.2*
*- new text that discusses the difference in relocation costs at lines 610-614*
*- new text that discusses the general aim of the FRISIA model at lines 898-921.*

**Treatment of relative SLR:** In line 224, the authors refer to 'relative SLR', but then immediately say they implement only "global mean SLR". At the same time, Table A1 suggests that some factor – whose derivation is not described anywhere – is used to localize the SLR projections. This should be clarified throughout.

We acknowledge that this part was not written clearly. In response to this comment and the comments from reviewer #2, we added a new section (now Sect. 3.2.2), which describes in one place how the local SLR is derived from the calculated global mean SLR by using the fingerprints from Slangen et al. (2014) (and then continue to explain how the effective surge height and coastal flooding are calculated). The ambiguity between "relative SLR" and the following sentence saying that we only model global mean SLR was resolved. In fact, the expected SLR also accounts for local SLR by a multiplication with a local SLR weight based on the above mentioned fingerprints.

*- new Section 3.2.2 clarifies the treatment of local and global SLR*
*- the ambiguous sentence located in Sect. 3.2.3 (l. 348) now reads:*

*"This relationship was derived using the model of GMSL presented in this manuscript, and it gives an estimate that is very close to the actually modelled GMSL rise in 50 years (Fig. A8). While this relationship was fitted using data of GMSL, the expected local SLR for each coastal zone can be calculated using the pre-computed weights for each coastal zone, which is provided as pre-processed input as described above. We note that this assumes that the relative contribution of the individual SLR processes stays approximately the same as in the projection used for the fit (SSP5-8.5). "*

**Declining GDP per capita:** It is a little hard to trace the relative effects of declining asset growth (decreasing GDP growth along coasts) and relocation (decreasing population) on GDP per capita (e.g., discussion around line 515). It would be helpful to look at this in greater detail.

We note that in this section we only look at the *No adaptation* strategy, in which relocation only occurs in response to people and assets becoming permanently inundated. The resulting relocation is therefore small in relative terms, and it happens for both assets and people, hence the effect on GDP per capita is mixed. This means that relocation has no major impact on the results in this section of the manuscript, and it is unclear to us how the discussion could be improved in this regard.

Nevertheless, in order to help with the interpretation of results, we have added another figure to the appendix, which shows time series of coastal population, storm damage resilience and the (reactive) retreat of people in response to inundation.

*- New Fig. A9 in appendix*

**Minor comments**

*Throughout:* The authors refer to timeseries of 'global SLR'; given that this is a univariate time series, I assume they mean 'global-mean SLR'.

We apologize for this inconsistency. We now introduce global mean sea level as GMSL and we either refer to GMSL when talking about the actual quantity of sea level change (mostly in sections 3.1 and 3.2.2, as well as in the Appendix) or SLR when talking about the general process. Additionally, we introduce a regional mean sea level (RMSL) rise as the driver of impacts for each coastal zone in Sect. 3.2.2, and we use local SLR, when we talking about the SLR for a segment.

*- Modified the manuscript to consistently make use of SLR or GMSL*

*Throughout*: The authors refer to "coastal GDP" and cite the SSP database, but do not explain where this coastal-specific GDP data comes from. To my knowledge, only country-level GDP estimates are contained in the SSP database. Is it the same population density-based downscaling approach used in Diaz, 2016?

Yes, this is indeed the same approach as in Diaz (2016). Based on this and other comments we have added a new section 3.2.1 which introduces the different inputs to the impacts and adaptation module of FRISIA in a dedicated place. We also discuss the use of this kind of input data in the new discussion section 7, that was added in response to other comments. We also now include a figure in the appendix that shows the evolution of the global coastal input data for GDP and population.

*- Text at lines 218-220 and 237-242 in new Sect. 3.2.1*
*- Discussion of our approach at lines 861-875 in Sect. 7*
*- New appendix figure A7*

*Line 222:* I am struggling to understand the reasoning behind building a second model for SLR, trained on the estimates from the first model. I assume this has something to do with the dynamic nature of the model, in which new SLR scenarios can be endogenously formed at each time step. However, this reasoning is not immediately clear in the manuscript. It would be helpful to include a sentence or two explaining why the outputs of the first SLR model cannot be used directly.

The reviewer is correct, a coupled version of FRISIA cannot rely on pre-computed time series that provide SLR in 50 years. Hence, a simple model to approximate SLR in 50 years had to be developed based solely on inputs that are available at that point of time in the model.

We added the following sentence at the beginning of the text that introduces this concept, which, together with the subsequent explanations, should make the reasoning more clear:

*- l. 345-347: "However, as the original purpose of FRISIA is to be run coupled to an IAM, which interactively provides the drivers for annual rates of GMSL rise, it will not always have access to pre-computed time series that provide the actually modelled GMSL rise in 50 years. Instead, we calculate the expected GMSL rise over the next 50 years, ..."*

*Line 994:* "MICI" is probably not an appropriate shorthand for factors driving high-end mass loss from Greenland; see Fox-Kemper et al. (2021) (...):

•In Greenland, stronger mass loss than currently projected might also occur (Aschwanden et al., 2019; Khan et al., 2020; T. Slater et al., 2020). For example, warming-induced dynamical changes in atmospheric circulation could enhance summer blocking and produce more frequent extreme melt events over Greenland similar to the record mass loss of more than 500 Gt in summer 2019 (Section 9.4.1.1; Delhasse et al., 2018; Sasgen et al., 2020). Cloud processes in polar areas that are not well represented in models could further enhance surface melt (Hofer et al., 2019), as could feedbacks between surface melt and the increasing albedo from meltwater, detritus and pigmented algae (Section 9.4.1.1; Cook et al., 2020). The same ice dynamical processes associated with basal melt and MISI discussed for Antarctica could also occur in Greenland, as long as the ice sheet is in contact with the ocean.

We agree that it is not technically correct to call the implemented threshold behaviour in the model "MICI", independent whether it is for the Greenland or the Antarctic ice sheet. We have decided to instead just generally speak of "uncertain high impact processes", independent of the ice sheet, as we should not refer to something more specific, given the simplicity of the implementation.

*- At several locations in the text, sentences that used "MICI" were rewritten as described above.*

**Code and Documentation**

I've verified the code can be cloned from the repo and run successfully. I've not otherwise examined the code. I have the following comments:

*The **README file** needs to provide adequate description to install code and run example. I suggest adding a requirements file. (Requires numby, pandas, matplotlib, netCDF4)*

We have added an environments.yml file to the repository and included information on how to set up the environment with conda in the README.

*In general, the README file is quite minimal, and I would argue insufficient to constitute the "user manual" expected to accompany GMD model description papers. The only documentation of the code is the two class-level doc strings in the source code.*

We have significantly expanded the README. It now includes a user guide that explains how to set up different types of model simulations, how to infer output from the model, how to run an uncertainty ensemble and how to activate additional feedback via model switches.

We also transferred some information from the doc strings to the README, but we prefer to not go into details of the implementation in the README. Instead, the preprint of this manuscript is linked there for anyone interested in more details.

**EXAMPLE_runFRISIA.py** *should produce output.*

We have expanded the script to produce a figure with example output.

*It would also be helpful to have **notebook(s) and/or scripts that fully replicate the results** shown in the text, especially Figs. 3–6.*

Notebooks to fully replicate the results presented in this manuscript can be found in the Zenodo repository that accompanies this submission (https://doi.org/10.5281/zenodo.15249065), which is also mentioned in the code and data availability statement.

Because of file size constraints for the full input (e.g. the timeseries of the 1001 member FaIR ensemble), and because the scripts are quite long and potentially more confusing than helpful for an inexperienced user, we decided to not include the scripts to recreate all figures from the manuscript directly in the GitHub repository, but we provide a link to the Zenodo at the start of the user manual in the README.

However, we acknowledge the fact that the creation of FRISIA input data for the different aggregation levels (i.e. global, bipolar and regional, as in the manuscript) is a relevant script to include in the GitHub repository. Therefore, we have reworked the respective script from the Zenodo to make it more generic and easy to use, even for creating completely new types of aggregation in FRISIA (still based on CIAM and DIVA data). This script was added to a new "processing" directory of the GitHub repository. Input data relevant to this script were also added to the repository.

**Anonymous Reviewer #2**

The authors present version 1 of the Feedback-based knowledge Repository for Integrated assessments of Sea level rise Impacts and Adaptation (FRISIAv1.0). FRISIA is designed for use in non-equilibrium Integrated Assessment Models (IAMs) and draws from published approaches for sea level as well as sea level impacts and adaptation modeling. FRISIA integrates dynamic feedbacks to more comprehensively capture economic damages from sea level rise impacts and has been coupled to and tested in a newly developed IAM framework.

The authors have to be commended for taking on a very complex task and successfully developing a tool that is able to estimate socio-economic damages from sea level rise in a transparent and plausible way. The main challenge for these tools is the development of reasonable simplifications and assumptions that allow for an efficient modeling chain while not neglecting necessary real-world boundary conditions. In this regard, I see a couple of issues that would need to be added and clarified. FRISIA will represent a much needed additional tool for estimating coastal damages from sea level rise in a research landscape that has been lacking diversity in modeling approaches.

We thank the reviewer for the overall very constructive feedback.

**Main comments:**

*Overview figure*

While the manuscript follows a logical structure it is hard for the reader to stay on top of the model structure given the number of components and assumptions introduced. In my view, it is necessary to include a schematic at the beginning of the model description that provides an overview of all FRISIA components and the coupling to FRIDA. This would establish a clear visual framework and it would be easier to link the following subsections to the overall modeling flow.

We have added a schematic (new Fig. 2) that shows the general structure of FRISIA, with a special focus on the inputs to and outputs from the model, as well as the internal flow of information within the model. Accordingly, we placed the figure at the beginning of the impacts and adaptation model description in Sect. 3.2.

*- new Figure 2*

*Adequate introduction of adopted assumptions and concepts*

Because FRISIA builds on existing modeling approaches, not all underlying assumptions and concepts are introduced in sufficient detail. This is particularly true for assumptions adopted from the CIAM and DIVA models. For example, FRISIA uses the DIVA coastal segments but it is never clarified what these

segments consist of. Table A1 includes information in this regard but it has to be clear upfront in the text which components constitute a coastal segment. Importantly, it is not clear that coastal segments exclude any classification according to coastal morphology which, of course, is key for modeling coastal flooding (see below). Another example is coastal assets. While "asset" is a common term in economic modeling, it is very important to be clear at the beginning of subsection 3.2.2 which coastal asset types are tracked by FRISIA.

We agree that more detail on the use of DIVA and CIAM assumptions would be useful. To that end and in response to other comments, we have added a new section 3.2.1 that discusses inputs to the impact and adaptation module and the use of DIVA data and CIAM formulations in general, as well as a new section 3.2.2 that discusses coastal flooding more specifically (especially in response to the comment below). We note, however, that DIVA does contain some information on coastal topography through the definition of area parameters, which we use to calculate inundated, generally susceptible and annually exposed asset and people fractions. This is introduced in the new section 3.2.2. We further added a new discussion section that elaborates on the use of DIVA or other coastal datasets, shortcomings of the model, the potential use of FRISIA, and avenues for future improvements.

Regarding the definition  of assets, neither the CIAM papers (Diaz, 2016; Wong et al. , 2022), nor the reference that is cited therein for the derivation of asset values (or capital, often used interchangeably) (Nordhaus, 2010), give a more specific definition of coastal assets. Hence we have built our model under the assumption that it is a standard economic definition, i.e. assets refer to the total value of past investments that produce economic output, after accounting for depreciation (Samuelson and Nordhaus, 2001). This includes, but is not limited to, residential and non-residential buildings, infrastructure, equipment and machinery. It should be noted that CIAM, and hence FRISIA, do not resolve anything more finely than at the segment level, using a constant density for the overall value of assets in a segment. No individual assets are modelled. We have put a similar explanation into the manuscript.

*- new Sect. 3.2., new Sect. 3.2.2, new Sect. 7*
*- new explanation at lines 379-383 in Sect. 3.2.4*

*Coastal flooding*

Arguably, the most important component of the FRISIA modeling chain is the projection of a reasonable aggregated coastal flooding indicator. This, however, is not captured with the adequate amount of detail. Even if a previously publish method (CIAM) is applied, it has to clear how coastal flooding is derived from the sea level rise projection which represent mean water level heights. To adequately model coastal flooding, three components are needed: sea level rise, extreme sea level or storm surge height, and wave height. The latter is crucial for coastal damage estimates but is neglected far too often. Including information from a wave model is not feasible for a light-weight impact model like FRISIA, of course. But this topic has to be resolved better. Would it be possible to add to the effective surge height parameter a term that would account for region-dependent typical wave heights during storm surges? I would suggest to include a new dedicated subsection 3.2.1 that would introduce the key concepts and discuss simplifications and caveats in detail.

We agree that the first version of the manuscript fell short of properly explaining how coastal flooding is modelled. We therefore follow the Reviewers' advice and have added a dedicated section (Sect. 3.2.2) on this that includes the equations for how coastal flooding is calculated on the DIVA segment level in a pre-processing step and how this information is then included into the FRISIA model. This section now also introduces the concept of the "effective flood height" as the driver of damages in FRISIA.

FRISIA indeed does not include the impact of local wave heights, and we agree that this is a shortcoming of the model. Adding a wave height term might very well be a feasible option in a future development step. However, we choose to not incorporate such a term in this manuscript already, because the focus here is on the base version of FRISIA that is comparable to CIAM (which does not account for this either) and the introduction of new feedbacks in FRISIA, following the advantages that specifically come with our new modelling philosophy. We added a sentence that our model does not include the impact of wave heights in the new section and in the new discussion Sect. 7.

*- new Sect. 3.2.2*
*- discussion of shortcomings/simplifications in Sect. 7*

*Shoreline types*

Similary and as already indicated before, the manuscript lacks information on how different shoreline morphologies are distinguished or not. Coastal flooding manifests very differently if a storm surge hits mangroves, sandy beaches, a steep rocky coast, or an urban coastal environment equipped with/without flood defenses. Again, even if the model cannot distinguish between all the relevant shoreline types, it is important to describe and discuss the underlying FRISIA assumptions. This should happen before the coastal protection subsection and could potentially be merged with the subsection clarifying how the effective surge height parameter is derived.

We agree and have incorporated a paragraph on this in the new Sect. 3.2.2, directly before the coastal protection subsection:

*"As a consequence of our highly aggregated modelling approach, our representation of coastal flooding is a major simplification of reality, as it does not incorporate any detailed information on coastal morphology, shoreline type or local wave heights, which would be necessary to explicitly model impacts of SLR at the local scale. This is a limitation of our modelling approach, which we discuss further in Sect. 7, but highlight here that the same simplifications are made in CIAM, which is more disaggregated than FRISIA, and that representations of SLR impacts in IAMs are typically far less complex."*

We also discuss the missing information on morphological data in the new discussion section (Sect. 7).

- text at lines 296-300 in Sect. 3.2.2
- text at lines 883-888 and 898-903 in Sect. 7

*Economic damages to coastal industries like fisheries and tourism*

In the compilation of cost formulations, I miss the description of costs that would incur from damages

to coastal economic sectors like fisheries or tourism. I believe that this is not covered under coastal assets per se. These costs or damages may be better captured in the feedback component as described in section 5, but it seems that they are also not explicitly captured there. As described, the feedbacks of reduced future investment, slower growth, reparation or failing to protect are insufficient to capture these damages, or they would at least have to be relabeled if such damages were captured via an existing GDP reduction term. Please elaborate on this matter and clarify.

Our model does not include these types of costs, just as the CIAM model does not include them. We added a few sentences on this at the end of the cost formulation section:

*"Furthermore, we note that, just like CIAM, our model does not capture the effect of SLR on economies that interact with the coastal environment, like fisheries, tourism or international trade. There is insufficient information available to appropriately calibrate and represent the indirect impacts of SLR on coastal industries. These costs are only partly represented via their relationship to coastal GDP and hence asset values, so that the calculated costs in FRISIAv1.0 are potentially underestimated."*

We also return to this matter in the new discussion section, where we highlight that, because of its focus on feedbacks, FRISIA is actually well-suited to include those impacts to coastal economic sectors in a future version. This, however, would require another study.

*- text at lines 562-566 in Sect. 4.1*
*- text at lines 892-896  and 930-932 in Sect. 7*

**Minor Comments:**

*L153,* It is unclear how the sea level component-wise "scale factors" are varied in the ensemble projections. Please clarify and also include a table in the Appendix that shows the parameters serving as "scale factors" for each sea level component and the established ranges.

We added more information in the text here, the full paragraph now reads as the following, with additional sentences in bold:

*"Given the large uncertainties in the parameter values, as well as the structure itself, we value simplicity over detail during the calibration process. Therefore, for each of the five components of GMSL rise, we define a "scale factor" that can be varied between 0 and 1, which respectively correspond to the minimum and maximum contribution of the individual components. **These factors are varied randomly and independently for each member of the FaIR input ensemble, so that the resulting time series of GMSL rise include uncertainty from both climate projections and SLR responses. Inside the SLR formulation for each component,** this factor is used to scale a set of one to four parameters within a predefined range simultaneously. **The calibrated range of the uncertainty parameters is given in the appendix Table A1.** While this approach does not cover the full parametric uncertainty, it makes sure that parameter combinations leading to unrealistic outcomes are avoided, and it is sufficient to cover the uncertainty ranges in AR6. These parameter ranges are defined during the calibration process and are described in appendix Sect. A."*

We also added the requested table in the Appendix.

*- New text at lines 161-170 in Sect. 3.1*
*- New table A1 in the appendix*

*L170,* This is, at the latest, where it has to be clarified what the DIVA coastal segments consist of, and that they don't really account for any morphological information, which one would intuitively assume.

We now have a dedicated section 3.2.1, which explains what kind of input is used from DIVA and other sources. Also the other new section 3.2.2 now comes back to further input originating from DIVA.

*- new Sect. 3.2.1*
*- new Sect. 3.2.2*

*Figure 1,* I suggest to use shading for the FRISIA projected ranges, slightly higher opacity for no-MICI and less opacity for MICI setups. Make all three panels equally wide and start in 1990, this should still allow for a readable legend.

An earlier version of this figure indeed used shadings, which we removed to be able to use .eps figures. We agree that shadings are more intuitive than the additional lines and hence went back to the previous version with shadings. However, we would like to stick with the current setup of panel (a) starting in 1900 and being wider than (b) and (c), in order to be able to also show the simulated GMSL curve for the 20th century once, as there might be future applications that will use FRISIA for this period as well, and because it shows that the more complex parametrisations for the Greenland and Antarctic ice sheets do not show unexpected behaviour for this period. Changes have been made for all corresponding figures from the Appendix as well.

*- Updated Figs. 1, A1, A2, A3, A4, A5, A6. Updated figure caption of Fig 1 and A1 to describe the shading.*

*L223,* Please explain why the 50-year time horizon is chosen, and not 30 years, for example.

*- added the following statement at lines 339-331 in Sect. 3.2.3:*

*"because, on the one hand, this gives enough time for building sufficient protection, while, on the other hand, there is still a relative small uncertainty in SLR projections over 50 years compared to the high uncertainty over longer time scales (see Fig. 1). In general, this follows the assumption that coastal zones, which adapt to SLR by building flood protection, will do so not just in response to experiencing changes in sea level, but using their knowledge about future SLR. "*

*L224,* It is not clear to me why it is necessary to derive the expected SLR in 50 years, SLR50, using GSAT and CO2 emissions. The sea level model delivers SLR50 until 2150 if we assume that the projections are produced until 2200, as shown in the sea level figures.

It is true that an offline version of FRISIA can rely on pre-computed time series of SLR. However, a version of FRISIA that is dynamically coupled to an IAM, which is the main application of our model, cannot rely on pre-computed time series that provide SLR in 50 years. For a given year, the SLR for subsequent years is calculated at later time steps of the model, which is why the SLR in 50 years cannot be taken from the internal sea level model. Hence, a simple model to approximate SLR in 50 years had to be developed based solely on inputs that are available at that point of time in the model.

We added the following sentence at the beginning of the text that introduces this concept, which, together with the subsequent explanations, should make the reasoning more clear:

*- l. 345-347: "However, as the original purpose of FRISIA is to be run coupled to an IAM, which interactively provides the drivers for annual rates of GMSL rise, it will not always have access to pre-computed time series that provide the actually modelled GMSL rise in 50 years. Instead, we calculate the expected GMSL rise over the next 50 years, ..."*

*L231,* The current effective flood height S will be much better understood after adequately introducing and explaining how it is derived (see above major comment).

Yes, we removed this sentence that shortly introduced the effective flood height here, as this is now explained in more detail in the previous section.

*L247,* Please describe here in more detail which assets types are tracked (see above major comment).

See response to major comment above.

*- new text at lines 379-383 in Sect.* 3.2.4

*L351,* Would more recently published information change the weights?

More recent information might change the numbers, but we do not expect a qualitative change in our results. We used Slangen et al. (2014) because the data was easily accessible and also used in the CIAM study that we use as reference in this manuscript. We are also not aware of any substantial change in newer data. We might still update also these weights in the future, as we also update the underlying database on the coastal segments. We now list this as a potential future improvement in the new discussion section.

*- line 930 in Sect. 7*

*L367-371,* This feels more like a paragraph for the discussion/conclusion.

Agreed, we have reworked this paragraph into the new discussion Sect. 7

*Figure 3,* Why do the surge fatality projections in panels a-c show a kink in 2100? Please explain and discuss in the text.

The first paragraph of section 4.2 discusses that GDP and population data are continued with constant values after 2100 (as in the reference CIAM simulations), but we did not explicitly mention that this leads to the visible kink in several time series. We have added the following sentence:

*- l. 572-573: "This continuation with constant population and GDP leads to a visible kink in the time series, and the form of data continuation needs to be considered when interpreting the results. "*

Furthermore, we now explain why fatalities start to increase faster after 2100 in the discussion of the results a few paragraphs below:

*- l. 620-629: "The time series of SLR-driven storm surge fatalities shows a visible kink and faster increase after 2100, while storm surge damages to assets do not show this behaviour. The reason for the kink is the continuation of input data with constant values as described above. But what is the*

*driver of the change in behaviour? The global coastal population is decreasing before 2100, while GDP increases. This leads to a decreasing number of people being potentially susceptible, and, at the same time, and increase of the resilience to damages due to a higher GDP per capita. Both trends counteract the increase of storm surge fatalities due to SLR. After 2100, these trends stop so that the remaining driver is the increase of storm surge fatalities from SLR. The difference for assets is that before 2100 damages are already accelerating, driven by SLR and the increase of asset values with GDP. After 2100 the continuation with constant values slows down the increase of asset values, but also storm surge resilience ceases to increase. Hence, there is no visible change in the evolution of storm surge damage to assets, despite a substantial change in the underlying socio-economic dynamics."*

*Figure 4*, There is not a sufficient description and explanation provided in the text why relocation costs under SSP5-8.5 show a peak in 2100 (panel f).

This is indeed an interesting behaviour. Discussing it requires some longer explanation, which we have added to the manuscript:

*"Interestingly, FRISIA simulates a peak of relocation costs under the Retreat strategy in 2100, but only for SSP5-8.5 (Fig. 5 (f)). Relocation costs are defined as the costs to move mobile assets and people, and the costs to demolish abandoned immobile assets. These costs are proportional to GDP (assets) and GDP per capita (people), both of which stop to increase after 2100, as a consequence of the chosen continuation of SSP data beyond that year (compare Fig. A7). This explains one part of why relocation costs stop growing after 2100, and also why the increase of flooding costs slows down in that year (Fig. 5 (i)). However, the question remains why relocation costs increase in SSP5-8.5 between approximately 2030 and 2100, while they remain stable or decrease in the other scenarios. One driver for this is that generally GDP increases a lot faster in SSP5-8.5, ending up being almost a factor of two higher in 2100 than in the other scenarios before considering retreat. It also keeps accelerating, while GDP grows more linearly in SSP2-4.5 and growth even decelerates in SSP1-2.6 (this is also a driver of the much faster increase of surge damages in SSP5-8.5 under no adaptation, Fig. 3 (d-f)). The final component to describe the peak shape in relocation costs is the much stronger SLR in SSP5-8.5, which drives the same shape, albeit in weaker form, also in relative terms for asset retreat (not shown). The insight here is that under a strong (expected) SLR all assets that are initially safe because of currently existing flood protection will become susceptible to storm surge damages in the course of the 21st century and hence retreat. This happens in a relatively short amount of time in SSP5-8.5, while it is more spread in the other scenarios due to slower SLR. The peak shape is therefore also a consequence of including the initial flood protection from the DIVA dataset even in our retreat strategy simulations, which has not been included in CIAM. Correspondingly, the counter-factual scenario of no initial flood protection (dotted lines), does not show the peak shape, as here assets and people retreat already in the first years of the simulation (at lower costs, but much stronger in relative terms)."*

*- added paragraph at lines 646-663 in Sect. 4.2.*

*L478*, "reasonable" may be a bit strong given the large number of assumptions, but the model does produce estimates that are comparable with other existing study results.

*- Removed the word "reasonable".*

*L560,* This is a very important finding which could be elevated more in the conclusion.

Agreed. Added the following text to the conclusion, after we mention the impact of the other newly added feedback:

*- lines 954-958 in Sect. 8: "Furthermore, we have added a limit on the amount of money available for raising flood protection, which is typically not included in other SLR impact models. Adding such a limit means that some coastal regions will fail to fully protect or even build new flood protection at all in scenarios of extreme SLR in the future. The potential failure to further invest into flood protection is an important feedback that should be considered in adaptation decisions."*

*L620-624,* I do not understand the seemingly arbitrary choice of 0.8 m (0.78 to 0.82 m) of global mean SLR in 2100, also because this would be closer to the more pessimistic AR6 projections under SSP5-8.5 than what could be considered "realistic" if a current policy interpretation was used (SSP2-4.5 would be closest then). If it is the intention to use SLR under very high emissions, then I would suggest to actually reference the medium confidence AR6 projected range of 0.77m (0.63 to 1.01 m) as target for the ensemble selection.

We agree that the choice of 0.8 m is somewhat arbitrary (the choice was indeed driven by the idea to use a scenario of relatively high SLR), and that the AR6 mean GMSL rise estimate for 2100 from SSP5-8.5 is a better value. We adapted the manuscript accordingly. The new reference value leads to more ensemble members being selected (1368 instead of 1178 out of 20,000) and slightly changed impact estimates that do not affect the conclusion. For example, the estimated range for the global GDP reduction in a scenario of no adaptation is now assumed to be 1.5 - 6.2 % from 1.6 - 6.1% (the range broadened a bit in both directions, because more ensemble members are selected, but generally there are now slightly smaller impacts).

*- Updated Figs. 8 (previously Fig.7) and A10 (previously Fig. A8).*
*- Updated the numbers discussed in sections 6, 8 (previously Sect. 7) and the abstract*
*- Updated the reasoning a bit:*

*-l. 823-827: "From this ensemble, we draw all the realisations that lead to approximately 0.77 (0.75-0.79) m of GMSL rise by 2100 (the median value of IPCC AR6 projections from SSP5-8.5), leaving us with 1368 samples. We choose the SSP5-8.5 GMSL rise because for the analysis in this section we want a substantial increase of GMSL, which is still reached by enough samples to adequately cover the spectrum of possible socio-economic developments in FRIDA"*

*L651,* In the conclusion, it is at least important to discuss in more detail the overall challenges of adequately estimating future damages from sea level rise, also because it is so hard to integrate non-economic damages in such assessments.

We agree that a deeper investigation of uncertainties and model shortcomings is useful. To this end, we have added a discussion section that now goes into the details of this. We prefer this over adding information to the conclusion section, because this way the topic can be fully discussed, while preserving a concise conclusion section for the paper. The new section covers a discussion of how

FRISIA generally handles uncertainty, as well as a discussion of model shortcomings and potential improvements for the future.

*- Added new Section 7.*

**Code availability and documentation**

I have cloned the FRISIA code repository and successfully ran the example scripts. It would be useful if the authors expanded the example script to also save key results in a dedicated output directory to increase usability.

We note that we have significantly updated the GitHub repository in terms of documentation, and included an additional script for input pre-processing there. Please see the response to Reviewer#1 for more details.

The EXAMPLE_runFRISIA.py script was expanded to now also produce a plot of some example data, similar to the other example script. The existing scripts, together with the new user manual in the README, should increase the usability of the model already a lot and make it easier for a user to produce their own output. Therefore we do not add an output directory that contains output time series, as it highly depends on the users interest what data is relevant. For the data used in this manuscript, the Zenodo repository contains all the relevant scripts and instructions to fully reproduce the results presented here.

References

Diaz, D. B.: Estimating global damages from sea level rise with the Coastal Impact and Adaptation Model (CIAM), Climatic Change, 137, 143–156, https://doi.org/10.1007/s10584-016-1675-4, 2016.

Wong, T. E., Ledna, C., Rennels, L., Sheets, H., Errickson, F. C., Diaz, D., and Anthoff, D.: Sea Level and Socioeconomic Uncertainty Drives High-End Coastal Adaptation Costs, Earth's Future, 10, e2022EF003 061, https://doi.org/10.1029/2022EF003061, 2022

Nordhaus, W. D.: The economic of Hurricanes and Implications of global warming, Climate Change Economics, 01, 1–20, https://doi.org/10.1142/S2010007810000054, 2010

Samuelson, P. and Nordhaus, W.: Economics, McGraw-Hill Higher Education, McGraw-Hill, ISBN 9780071180641, https://books.google.de/books?id=jqkrAAAAYAAJ, 2001.

Slangen, A.B.A., Carson, M., Katsman, C.A. *et al.* Projecting twenty-first century regional sea-level changes. *Climatic Change* **124**, 317–332, https://doi.org/10.1007/s10584-014-1080-9, 2014.